# NEAR OPTIMAL PRIVATE AND ROBUST LINEAR REGRESSION

## ABSTRACT

We study the canonical statistical estimation problem of linear regression from $n$ i.i.d. examples under $(\varepsilon, \delta)$-differential privacy when a fraction of response variables are adversarially corrupted. We propose a variant of the popular differentially private stochastic gradient descent (DP-SGD) algorithm with two innovations: a full-batch gradient descent to improve sample complexity and a novel adaptive clipping to guarantee robustness. When there is no adversarial corruption, this algorithm improves upon the existing state-of-the-art approach and achieves near optimal sample complexity. Under label-corruption, this is the first efficient linear regression algorithm to provably guarantee both $(\epsilon, \delta)$-DP and robustness. Synthetic experiments confirm the superiority of our approach.

## 1 INTRODUCTION

Differential Privacy (DP) is a widely accepted notion of privacy introduced in (Dwork et al., 2006), which is now standard in industry and government (Tang et al., 2017; Erlingsson et al., 2014; Fanti et al., 2016; Abowd, 2018). A query to a database is said to be $(\varepsilon, \delta)$-differentially private if a strong adversary who knows all other entries cannot identify with high confidence whether you participated in the database or not. The parameters $\varepsilon$ and $\delta$ restrict the Type-I and Type-II errors achievable by the adversary (Kairouz et al., 2015). Smaller $\varepsilon > 0$ and $\delta \in [0, 1]$ imply stronger privacy guarantees.

Significant advances have been made recently in understanding the utility-privacy trade-offs in canonical statistical tasks. We provide a survey in App. A. However, several important questions remain open, some of which we address below. A canonical statistical task of linear regression is when $n$ i.i.d. samples, $\{(x_i \in \mathbb{R}^d, y_i \in \mathbb{R})\}_{i=1}^n$, are drawn from $x_i \sim \mathcal{N}(0, \Sigma)$, $y_i = x_i^\top w^* + z_i$, and $z_i \sim \mathcal{N}(0, \sigma^2)$. The error is measured in $\|\hat{w} - w^*\|_\Sigma := \|\Sigma^{1/2}(\hat{w} - w^*)\|$, which correctly accounts for the signal-to-noise ratio in each direction; in the direction of large eigenvalue of $\Sigma$, we have larger signal in $x_i$ and the same noise in $z_i$, and hence expect smaller error.

When computational complexity is not concerned, the best known algorithm is introduced by Liu et al. (2022b), called High-dimensional Propose-Test-Release (HPTR), that can be flexibly applied to a variety of statistical tasks to achieve the optimal sample complexity under $(\varepsilon, \delta)$-DP. For linear regression, $n = O(d/\alpha^2 + d/(\varepsilon\alpha))$ samples are sufficient for HPTR to achieve an error of $(1/\sigma)\|\hat{w} - w^*\|_\Sigma^2 = \alpha$ with high probability. This is optimal, matching known information theoretic lower bounds. It remains an important open question if this can be achieved with an efficient algorithm. After a series of work surveyed in App. A, Varshney et al. (2022) achieves the best known sample complexity for an efficient algorithm: $n = \tilde{O}(\kappa^2 d/\varepsilon + d/\alpha^2 + \kappa d/(\varepsilon\alpha))$. The last term is suboptimal by a factor of $\kappa$, the condition number of the covariance $\Sigma$ of the covariates, and the first term is unnecessary. We further close this gap in the following.

**Theorem 1** (informal version of Theorem 3 with no adversary). *Under the $(\Sigma, \sigma^2, w^*, K, a)$-model in Assumption 1, $n = \tilde{O}(d/\alpha^2 + \kappa^{1/2}d/(\varepsilon\alpha))$ samples are sufficient for Algorithm 1 to achieve an error rate of $(1/\sigma)\|\hat{w} - w^*\|_\Sigma^2 = \tilde{O}(\alpha)$ and $(\varepsilon, \delta)$-DP, where $\kappa := \lambda_{\max}(\Sigma)/\lambda_{\min}(\Sigma)$.*

Perhaps surprisingly, we show that the same algorithm is also robust against label-corruption, where an adversary selects arbitrary $\alpha_{\text{corrupt}}$ fraction of the data points and changes their response variables arbitrarily. When computational complexity is not concerned, the best known algorithm is HPTR by Liu et al. (2022b) that also provides optimal robustness and $(\varepsilon, \delta)$-DP simultaneously, i.e., $n =$

$O(d/\alpha^2 + d/(\varepsilon\alpha))$ samples are sufficient for HPTR to achieve an error of $(1/\sigma)\|\hat{w} - w^*\|_\Sigma^2 = \alpha$ for any corruption bounded by $\alpha_{\mathrm{corrupt}} \leq \alpha$. Note that this is a strong adversary who can corrupt both the covariate, $x_i$, and the response variable, $y_i$. Currently, there is no efficient algorithm that can guarantee both privacy and robustness for linear regression. Under a weaker adversary who can only corrupt the response variable, we close this gap in the following.

**Theorem 2** (informal version of Theorem 3 with adversarial label corruption). *Under the hypotheses of Theorem 1 and under $\alpha_{\mathrm{corrupt}}$-corruption model of Assumption 2, if $\alpha_{\mathrm{corrupt}} \leq \alpha$ then $n = \tilde{O}(d/\alpha^2 + \kappa^{1/2}d/(\varepsilon\alpha))$ samples are sufficient for Algorithm 1 to achieve an error rate of $(1/\sigma)\|\hat{w} - w^*\|_\Sigma^2 = \tilde{O}(\alpha)$ and $(\varepsilon, \delta)$-DP, where $\kappa := \lambda_{\max}(\Sigma)/\lambda_{\min}(\Sigma)$.*

We start with the formal description of the setting in Sec. 2 and present the approach of Varshney et al. (2022) for a private but non-robust linear regression. We build upon this approach and make two innovations. First, we propose full-batch gradient descent, which is more challenging to analyze but achieves improved dependence on the condition number $\kappa$. Crucial in overcoming the challenges in the analysis is the notion of *resilience* explained in our proof sketch (Sec. 6). Secondly, we propose novel adaptive clipping method that ensures robustness against label corruption. We present our main algorithm (Alg. 1) in Sec. 3 with theoretical analysis and justification of the assumptions. Our adaptive clipping is both robust and private. We use truncated mean to ensure robustness and private histogram to ensure privacy in Sec. 4. We present numerical experiments on synthetic data that demonstrates the sample efficiency of our approach in Sec. 5. We end with a sketch of our main proof ideas in Sec. 6, which might be of independent interest to those requiring tight analysis of linear regression in other settings.

## 2 PROBLEM FORMULATION AND BACKGROUND

For linear regression without adversarial corruption, the following assumption is standard for the uncorrupted dataset $S_{\mathrm{good}}$, except for the fact that we assume a more general family of $(K, a)$-sub-Weibull distributions that recovers the standard sub-Gaussian family as a special case when $a = 1/2$.

**Assumption 1** (($\Sigma, \sigma^2, w^*, K, a$)-model). *A multiset $S_{\mathrm{good}} = \{(x_i \in \mathbb{R}^d, y_i \in \mathbb{R})\}_{i=1}^n$ of $n$ i.i.d. samples is from a linear model $y_i = \langle x_i, w^* \rangle + z_i$, where the input vector $x_i$ is zero mean, $\mathbb{E}[x_i] = 0$, with a positive definite covariance $\Sigma := \mathbb{E}[x_i x_i^\top] \succ 0$, and the (input dependent) label noise $z_i$ is zero mean, $\mathbb{E}[z_i] = 0$, with variance $\sigma^2 := \mathbb{E}[z_i^2]$. We further assume $\mathbb{E}[x_i z_i] = 0$, which is equivalent to assuming that the true parameter $w^* = \Sigma^{-1}\mathbb{E}[y_i x_i]$. We assume that the marginal distribution of $x_i$ is $(K, a)$-sub-Weibull and that of $z_i$ is also $(K, a)$-sub-Weibull, as defined below.*

Sub-Weibull distributions provide Gaussian-like tail bounds determining the resilience of the dataset in Lemma J.7, which our analysis critically relies on and whose necessity is justified in Sec. 3.3.

**Definition 2.1** (sub-Weibull distribution Kuchibhotla & Chakrabortty (2018) ). *For some $K, a > 0$, we say a random vector $x \in \mathbb{R}^d$ is from a $(K, a)$-sub-Weibull distribution if for all $v \in \mathbb{R}^d$,*

$$\mathbb{E}\left[\exp\left(\left(\frac{\langle v, x\rangle^2}{K^2\mathbb{E}[\langle v, x\rangle^2]}\right)^{1/(2a)}\right)\right] \leq 1 \,.$$

Our goal is to estimate the unknown parameter $w^*$, given upper bounds on the sub-Weibull parameters $(K, a)$ and a corrupted dataset under the the standard definition of *label corruption* in (Bhatia et al., 2015). There are variations in literature, which we survey in Appendix A.

**Assumption 2** ($\alpha_{\mathrm{corrupt}}$-corruption). *Given a dataset $S_{\mathrm{good}} = \{(x_i, y_i)\}_{i=1}^n$, an adversary inspects all the data points, selects $\alpha_{\mathrm{corrupt}}n$ data points denoted as $S_r$, and replaces the labels with arbitrary labels while keeping the covariates unchanged. We let $S_{\mathrm{bad}}$ denote this set of $\alpha_{\mathrm{corrupt}}n$ newly labelled examples by the adversary. Let the resulting set be $S := S_{\mathrm{good}} \cup S_{\mathrm{bad}} \setminus S_r$. We further assume that the corruption rate is bounded by $\alpha_{\mathrm{corrupt}} \leq \bar{\alpha}$, where $\bar{\alpha}$ is a known positive constant satisfying $\bar{\alpha} \leq 1/10$, $72C_2 K^2 \bar{\alpha} \log^{2a}(1/(6\bar{\alpha})) \log(\kappa) \leq 1/2$, and $2C_2 K^2 \log^{2a}(1/(2\bar{\alpha})) \geq 1$ for the $(K, a)$-sub-Weibull distribution of interest and a positive constant $C_2$ defined in Lemma J.7 that only depends on $(K, a)$.*

**Notations.** A vector $x \in \mathbb{R}^d$ has the Euclidean norm $\|x\|$. For a matrix $M$, we use $\|M\|_2$ to denote the spectral norm. The error is measured in $\|\hat{w} - w^*\|_\Sigma := \|\Sigma^{1/2}(\hat{w} - w^*)\|$ for some PSD matrix $\Sigma$.

The identity matrix is denoted by $\mathbf{I}_d \in \mathbb{R}^{d \times d}$. Let $[n] = \{1, 2, \ldots, n\}$. $\tilde{O}(\cdot)$ hides some constants terms, $K, a = \Theta(1)$, and poly-logarithmic terms in $n$, $d$, $1/\varepsilon$, $\log(1/\delta)$, $1/\zeta$, and $1/\alpha_{\text{corrupt}}$.

## 2.1 BACKGROUND ON DP

Differential Privacy (DP) is a standard measure of privacy leakage when a dataset is accessed via queries, introduced by Dwork et al. (2006). Algorithms with strong DP guarantees provide plausible deniability to a strong adversary who knows all other entries in that dataset and tries to identify a particular user's entry (Kairouz et al., 2015). Two datasets $S$ and $S'$ are said to be neighbors if they differ at most by one entry, which is denoted by $S \sim S'$. A stochastic query $q$ is said to be $(\varepsilon, \delta)$-differentially private for some $\varepsilon > 0$ and $\delta \in [0, 1]$, if $\mathbb{P}(q(S) \in A) \leq e^{\varepsilon} \mathbb{P}(q(S) \in A) + \delta$, for all neighboring datasets $S \sim S'$ and all subset $A$ of the range of the query. We build upon two widely used DP primitives, the Gaussian mechanism and the private histogram. A central concept in DP mechanism design is the *sensitivity* of a query, defined as $\Delta_q := \sup_{S \sim S'} \|q(S) - q(S')\|$. We describe private histogram in App. B.

**Lemma 2.2** (Gaussian mechanism Dwork & Roth (2014)). *For a query $q$ with sensitivity $\Delta_q$, the Gaussian mechanism outputs $q(S) + \mathcal{N}(0, (\Delta_q \sqrt{2 \log(1.25/\delta)}/\varepsilon)^2 \mathbf{I}_d)$ and achieves $(\varepsilon, \delta)$-DP.*

## 2.2 STANDARD APPROACH IN PRIVATE LINEAR REGRESSION

When there is no adversarial corruption, the state-of-the-art approach introduced by Varshney et al. (2022) is based on stochastic gradient descent with clipping and additive Gaussian noise to ensure privacy. There are two main components in this approach: adaptive clipping and streaming SGD. Adaptive clipping with an appropriate threshold $\theta_t$ ensures that no data point is clipped while providing a bound on the sensitivity of the average mini-batch gradient, which ensures we do not add too much noise. The streaming approach, where a data point is only touched once and discarded, ensures independence between the past iterate $w_{t-1}$ and the gradients at round $t$, which the analysis critically relies on. For $T = \tilde{\Theta}(\kappa)$, iterations where $\kappa$ is the condition number of the covariance $\Sigma$ of the covariates, the dataset $S = \{(x_i, y_i)\}_{i=1}^n$ is partitioned into $\{B_t\}_{t=1}^T$ subsets of equal size. At each round $t < T$, the gradients are clipped and averaged with additive Gaussian noise:

$$w_{t+1} \leftarrow w_t - \eta \Big( \frac{1}{|B_t|} \sum_{i \in B_t} \text{clip}_{\theta_t}(x_i(w_t^\top x_i - y_i)) + \frac{\theta_t \sqrt{2 \log(1.25/\delta)}}{\varepsilon |B_t|} \nu_t \Big), \tag{1}$$

where $\nu_t \sim \mathcal{N}(0, \mathbf{I}_d)$. In Varshney et al. (2022), a variation of this streaming SGD is shown to require $n = \tilde{O}(\kappa^2 d/\varepsilon + d/\alpha^2 + \kappa d/(\varepsilon \alpha))$ to achieve an error of $\|w_T - w^*\|_\Sigma^2 = O(\sigma^2 \alpha^2)$.

Our approach builds upon such gradient based methods but makes two important innovations. First, we use full-batch gradient descent, as opposed to the streaming SGD above. Using all $n$ samples reduces the sensitivity of the per-round gradient average, allowing us to improve the sample complexity to $n = \tilde{O}(d/\alpha^2 + \kappa^{1/2} d/(\varepsilon \alpha))$ to achieve an error of $\|w_T - w^*\|_\Sigma^2 = O(\sigma^2 \alpha^2)$. However, we lose the independence between $w_{t-1}$ and the gradients in the current round, which makes the analysis more challenging. We instead rely on *resilience* to precisely track the bias and variance of the (dependent) full-batch gradient average. Resilience is a central concept in robust statistics which we explain in Sec. 6. The second innovation we make is separately clipping $x_i$ and $(w_t^\top x_i - y_i)$ in the gradient. This is critical in achieving robustness to label-corruption, as we explain in Sec. 3.1.

# 3 ROBUST AND DIFFERENTIALLY PRIVATE LINEAR REGRESSION

We introduce a gradient descent approach for linear regression with a novel adaptive clipping that ensures robustness against label-corruption. This achieves a near-optimal sample complexity and, for the special case of private linear regression without adversarial corruption, improves upon the state-of-the-art algorithm.

## 3.1 ALGORITHM

The skeleton of our approach in Alg. 1 is the general DP-SGD Abadi et al. (2016); Song et al. (2013) with adaptive clipping Andrew et al. (2021). However, the standard adaptive clipping is not robust

against label-corruption under the more general $(K, a)$-sub-Weibull assumption. In particular, it is possible under sub-Weibull distribution that a positive fraction of the covariates are close to the origin, which is not possible under Gaussian data due to concentration. In this case, the adversary can select to corrupt those points with small norm, $\|x_i\|$, making large changes in the residual, $(y_i - w_t^\top x_i)$, while evading the standard clipping (by the norm of the gradient), since the norm of the gradient, $\|x_i(y_i - w_t^\top x_i)\| = \|x_i\| |y_i - w_t^\top x_i|$, can remain under the threshold. This is problematic, since the bias due to the corrupted samples in the gradient scales proportional to the magnitude of the residual (after clipping). To this end, we propose clipping the norm and the residual separately: $\text{clip}_\Theta(x_i)\text{clip}_{\theta_t}\left(w_t^\top x_i - y_i\right)$. This keeps the sensitivity of gradient average bounded by $\Theta\theta_t$, and the subsequent Gaussian mechanism in line 8 ensures $(\varepsilon_0, \delta_0)$-DP at each round. Applying advanced composition in Lemma B.4 of $T$ rounds, this ensures end-to-end $(\varepsilon, \delta)$-DP.

**Novel adaptive clipping.** In $\text{clip}_\Theta(x_i)$, the only purpose of clipping the covariate by its norm, $\|x_i\|$, is to bound the sensitivity of the resulting clipped gradient. In particular, we do not need to make it robust as there is no corruption in the covariates. Ideally, we want to select the smallest threshold $\Theta$ that does not clip any of the covariates. Since the norm of a covariate is upper bounded by $\|x_i\|^2 \leq K^2 \text{Tr}(\Sigma) \log^{2a}(1/\zeta)$ with probability $1 - \zeta$ (Lemma J.3), we estimate the unknown $\text{Tr}(\Sigma)$ using Private Norm Estimator in Alg. 3 in App. F and set the norm threshold $\Theta = K\sqrt{2\Gamma} \log^a(n/\zeta)$ (line 3). The $n$ in the logarithm ensures that the union bound holds.

In $\text{clip}_{\theta_t}(w_t^\top x_i - y_i)$, the purpose of clipping the residual by its magnitude, $|y_i - w_t^\top x_i| = |(w^* - w_t)^\top x_i + z_i|$, is to bound the sensitivity of the gradient and also to provide robustness against label-corruption. We want to choose a threshold that only clips corrupt data points and at most a few clean data points. We know that any set of $(1 - 2\alpha_{\text{corrupts}})$ fraction of the clean data points is sufficient to get a good estimate of the average gradient, and we can find such a large enough set of points that satisfy $|(w^* - w_t)^\top x_i + z_i|^2 \leq (\|w_t - w^*\|_\Sigma^2 + \sigma^2)CK^2 \log^{2a}(1/(2\alpha))$ from Lemma J.3. At the same time, this threshold on the residual is small enough to guarantee robustness against the label-corrupted samples. We introduce Robust Private Distance Estimator in Alg. 2 to estimate the unknown (squared and shifted) distance, $\|w_t - w^*\|_\Sigma^2 + \sigma^2$, and set the distance threshold $\theta_t = 2\sqrt{2\gamma_t}\sqrt{9C_2K^2 \log^{2a}(1/(2\alpha))}$ (line 6). Both norm and distance estimation rely on private histogram (Lemma B.1), but over a set of statistics computed on partitioned datasets, which we explain in detail in Sec. 4.

---

**Algorithm 1:** Robust and Private Linear Regression

**Input:** dataset $S = \{(x_i, y_i)\}_{i=1}^{3n}$, privacy parameters $(\varepsilon, \delta)$, number of iterations $T$, learning rate $\eta$, failure probability $\zeta$, target error rate $\alpha$, distribution parameter $(K, a)$

1. Partition dataset $S$ into three equal sized disjoint subsets $S = S_1 \cup S_2 \cup S_3$.
2. $\delta_0 \leftarrow \delta/(2T)$, $\varepsilon_0 \leftarrow \varepsilon/(4\sqrt{T \log(1/\delta_0)})$, $\zeta_0 \leftarrow \zeta/3$, $w_0 \leftarrow 0$
3. $\Gamma \leftarrow \text{PrivateNormEstimator}(S_1, \varepsilon_0, \delta_0, \zeta_0)$, $\Theta \leftarrow K\sqrt{2\Gamma} \log^a(n/\zeta_0)$
4. **for** $t = 1, 2, \ldots, T-1$ **do**
5. $\quad$ $\gamma_t \leftarrow \text{RobustPrivateDistanceEstimator}(S_2, w_t, \varepsilon_0, \delta_0, \alpha, \zeta_0)$
6. $\quad$ $\theta_t \leftarrow 2\sqrt{2\gamma_t} \cdot \sqrt{9C_2K^2 \log^{2a}(1/(2\alpha))}$.
7. $\quad$ Sample $\nu_t \sim \mathcal{N}(0, \mathbf{I}_d)$
8. $\quad$ $w_{t+1} \leftarrow w_t - \eta\left(\frac{1}{n}\sum_{i \in S_3}\left(\text{clip}_\Theta(x_i)\text{clip}_{\theta_t}\left(w_t^\top x_i - y_i\right)\right) + \frac{\sqrt{2\log(1.25/\delta_0)}\Theta\theta_t}{\varepsilon_0 n} \cdot \nu_t\right)$
9. Return $w_T$

---

### 3.2 ANALYSIS

We show that Algorithm 1 achieves a near-optimal sample complexity. We provide a proof in Appendix H and a sketch of the proof in Section 6. We address the necessity of the assumptions in Sec. 3.3, along with some lower bounds.

**Theorem 3.** *Algorithm 1 is $(\varepsilon, \delta)$-DP. Under $(\Sigma, \sigma^2, w^*, K, a)$-model of Assumption 1 and $\alpha_{\text{corrupt}}$-corruption of Assumption 2 and for any failure probability $\zeta \in (0, 1)$ and target error rate $\alpha \geq$*

$\alpha_{\text{corrupt}}$, *if sample size is large enough such that*

$$n = \tilde{O}\left(K^2 d \log^{2a+1}\left(\frac{1}{\zeta}\right) + \frac{d + \log(1/\zeta)}{\alpha^2} + \frac{d T^{1/2} \log(\frac{1}{\delta}) \log^a(\frac{1}{\zeta})}{\varepsilon \alpha}\right), \tag{2}$$

*with a large enough constant where $\tilde{O}$ hides poly-logarithmic terms in $d$, $n$, and $\kappa$, then the choices of a small enough step size, $\eta \leq 1/(1.1\lambda_{\max}(\Sigma))$, and the number of iterations, $T = \tilde{\Theta}(\kappa \log(\|w^*\|))$ for a condition number of the covariance $\kappa := \lambda_{\max}(\Sigma)/\lambda_{\min}(\Sigma)$, ensures that, with probability $1 - \zeta$, Algorithm 1 achieves*

$$\mathbb{E}_{\nu_1,\cdots,\nu_t \sim \mathcal{N}(0, \mathbf{I}_d)}\left[\|w_T - w^*\|_\Sigma^2\right] = \tilde{O}\left(K^4 \sigma^2 \alpha^2 \log^{4a}\left(\frac{1}{\alpha}\right)\right), \tag{3}$$

*where the expectation is taken over the noise added for DP, and $\tilde{\Theta}(\cdot)$ hides logarithmic terms in $K, \sigma, d, n, 1/\varepsilon, \log(1/\delta), 1/\alpha$, and $\kappa$.*

**Optimality.** Omitting some constant and logarithmic terms, Alg. 1 requires

$$n = \tilde{O}\left(\frac{d}{\alpha^2} + \frac{\kappa^{1/2} d \log(1/\delta)}{\varepsilon \alpha}\right), \tag{4}$$

samples to ensure an error rate of $\mathbb{E}[\|w_T - w^*\|_\Sigma^2] = \tilde{O}(\sigma^2 \alpha^2)$ for any $\alpha \geq \alpha_{\text{corrupt}}$. The lower bound on the achievable error of $\sigma^2 \alpha^2 \geq \sigma^2 \alpha_{\text{corrupt}}^2$ is due to the label-corruption and cannot be improved, as it matches an information theoretic lower bound we provide in Proposition 3.1. In the special case when the covariate follows a sub-Gaussian distribution, that is $(K, 1/2)$-sub-Weibull for a constant $K$, there is an $n = \Omega(d/\alpha^2 + d/(\varepsilon\alpha))$ lower bound (Cai et al. (2019), Theorem 4.1), and our upper bound matches this lower bound up to a factor of $\kappa^{1/2}$ in the second term and other logarithmic factors. Eq. (4) is the best known rate among all efficient private linear regression algorithms, strictly improving upon existing methods when $\log(1/\delta) = \tilde{O}(1)$. We discuss some exponential time algorithms that closes the $\kappa^{1/2}$ gap in Sec. 3.3.

**Comparisons with the state-of-the-art.** The best existing efficient algorithm by Varshney et al. (2022) can only handle the case where there is *no adversarial corruption*, and requires $n = \tilde{O}(\kappa^2 d\sqrt{\log(1/\delta)}/\varepsilon + d/\alpha^2 + \kappa d\sqrt{\log(1/\delta)}/(\varepsilon\alpha))$ to achieve an error rate of $\sigma^2 \alpha^2$. Compared to Eq. (4), the first term dominates in its dependence in $\kappa$, which is a factor of $\kappa$ larger than Eq. (4). The third term is larger by a factor of $\kappa^{1/2}$ but smaller by a factor of $\log^{1/2}(1/\delta)$, compared to the second term in Eq. (4).

In the *non-private case*, when $\varepsilon = \infty$, a recent line of work has developed algorithms for linear regression that are robust to label corruptions (Bhatia et al., 2015; 2017; Suggala et al., 2019; Dalalyan & Thompson, 2019). Of these, Bhatia et al. (2015); Dalalyan & Thompson (2019) are relevant to our work as they consider the same adversary model as us. When $x_i$'s and $z_i$'s are sampled from $\mathcal{N}(0, \Sigma)$ and $\mathcal{N}(0, \sigma^2)$, Dalalyan & Thompson (2019) proposed a Huber loss based estimator that achieves error rate of $\sigma^2 \alpha^2 \log^2(n/\delta)$ when $n = \tilde{O}(\kappa^2 d/\alpha^2)$. Under the same setting, Bhatia et al. (2015) proposed a hard thresholding based estimator that achieves $\sigma^2 \alpha^2$ error rate with $\tilde{O}(d/\alpha^2)$ sample complexity. Our results in Theorem 3 match these rates, except for the sub-optimal dependence on $\log^{4a}(1/\alpha)$. Another line of work considered both label and covariate corruptions and developed optimal algorithms for parameter recovery (Diakonikolas et al., 2019c;b; Prasad et al., 2018; Pensia et al., 2020; Cherapanamjeri et al., 2020; Jambulapati et al., 2020; Klivans et al., 2018; Bakshi & Prasad, 2021; Zhu et al., 2019; Depersin, 2020). The best existing efficient algorithm , e.g. Pensia et al. (2020) achieves error rate of $\sigma^2 \alpha^2 \log(1/\alpha)$ when $n = \tilde{O}(d/\alpha^2)$, and the uncorrupted $x_i$ and $z_i$ are sampled from $\mathcal{N}(0, I)$ and $\mathcal{N}(0, \sigma^2)$.

Under both privacy requirements and adversarial corruption, the only algorithm with a provable guarantee is the exponential time approach, known as High-dimensional Propose-Test-Release (HPTR), of (Liu et al., 2022b, Corollary C.2), which achieves a sample complexity of $n = O(d/\alpha^2 + (d + \log(1/\delta))/(\varepsilon\alpha))$. Notice that there is no dependence on $\kappa$ and the $\log(1/\delta)$ term scales as $1/(\varepsilon\alpha)$ as opposed to $\kappa d^{1/2}/(\varepsilon\alpha)$ of Eq. (4). It remains an open question if *computationally efficient* private linear regression algorithms can achieve such a $\kappa$-independent sample complexity. Further, HPTR is robust against a stronger adversary who corrupts the covariates also and not just the labels. Under this more powerful adversary, it remains an open question if there is an efficient algorithm that achieves $n = O(d/\alpha^2 + d/(\varepsilon\alpha))$ sample complexity even for constant $\kappa$ and $\delta$.

### 3.3 LOWER BOUNDS

**Necessity of our assumptions.** A tail assumption on the covariate $x_i$ such as Assumption 1 is necessary to achieve $n = O(d)$ sample complexity in Eq. (4). Even when the covariance $\Sigma$ is close to identity, without further assumptions on the tail of covariate $x$, the result in Bassily et al. (2014) implies that for $\delta < 1/n$ and sufficiently large $n$, no $(\varepsilon, \delta)$-DP estimator can achieve excess risk $\|\hat{w} - w^*\|_\Sigma^2$ better than $\Omega(d^3/(\varepsilon^2 n^2))$ (see Eq. (3) in Wang (2018)). Note that this lower bound is a factor $d$ larger than our upper bound that benefits from the additional tail assumption.

A tail assumption on the noise $z_i$ such as Assumption 1 is necessary to achieve $n = O(d/(\varepsilon\alpha))$ dependence on the sample complexity in Eq. (4). For heavy-tailed noise, such as $k$-th moment bounded noise, the dependence can be significantly larger. (Liu et al., 2022b, Proposition C.5) implies that for $\delta = e^{-\Theta(d)}$ and 4-th moment bounded $x_i$ and $z_i$, any $(\varepsilon, \delta)$-DP estimator requires $n = \Omega(d/(\varepsilon\alpha^2))$ to achieve excess risk $\mathbb{E}[\|\hat{w} - w^*\|_\Sigma^2] = \tilde{O}(\sigma^2\alpha^2)$.

The assumption that only label is corrupted is critical for Algorithm 1. The average of the (adaptively) clipped gradient can be significantly more biased, if the adversary can place the covariates of the corrupted samples in the same direction. In particular, the bound on the bias of our gradient step in Eq. (42) in App. H would no longer hold. Against such strong attacks, one requires additional steps to estimate the mean of the gradients robustly and privately, similar to those used in robust private mean estimation Liu et al. (2021); Kothari et al. (2021); Hopkins et al. (2022); Ashtiani & Liaw (2022). Pursuing this direction is outside the scope of this paper.

**Lower bounds under label corruption.** Under the $\alpha_{\mathrm{corrupt}}$ label corruption setting (Assumption 2), even with infinite data and without privacy constraints, no algorithm is able to learn $w^*$ with $\ell_2$ error better than $\alpha_{\mathrm{corrupt}}$. We provide a formal derivation for completeness.

**Proposition 3.1.** *Let $\mathcal{D}_{\Sigma,\sigma^2,w^*,K,a}$ be a class of joint distributions on $(x_i, y_i)$ from $(\Sigma, \sigma^2, w^*, K, a)$-model in Assumption 1. Let $S_{n,\alpha}$ be an $\alpha$-corrupted dataset of $n$ i.i.d. samples from some distribution $\mathcal{D} \in \mathcal{D}_{\Sigma,\sigma^2,w^*,K,a}$ under Assumption 2. Let $\mathcal{M}$ be a class of estimators that are functions over the datasets $S_{n,\alpha}$. Then there exists a positive constant $c$ such that*

$$\min_{n,\hat{w}\in\mathcal{M}} \max_{S_{n,\alpha},\mathcal{D}\in\mathcal{D}_{\Sigma,\sigma^2,w^*,K,a},w^*,K,a,} \mathbb{E}[\|\hat{w} - w^*\|_\Sigma^2] \geq c\,\alpha^2\,\sigma^2 \,. \tag{5}$$

A proof is provided in Appendix I.1. A similar lower bound can be found in (Bakshi & Prasad, 2021, Theorem 6.1).

## 4 ADAPTIVE CLIPPING FOR THE GRADIENT NORM

In the ideal clipping thresholds for norm and the residual, there is an unknown terms which we need to estimate adaptively, $(\|w_t - w^*\|_\Sigma^2 + \sigma^2)$ and $\mathrm{Tr}(\Sigma)$, up to a constant multiplicative error. We privately estimate the (squared and shifted) distance to optimum, $(\|w_t - w^*\|_\Sigma^2 + \sigma^2)$, with Alg. 2 and privately estimate the average input norm, $\mathbb{E}[\|x_i\|^2] = \mathrm{Tr}(\Sigma)$, with Alg. 3 in App. F. These are used to get the clipping thresholds in Alg. 1. We propose a trimmed mean approach below for distance estimation. The norm estimator is similar and is provided in App. F.

**Private distance estimation using private trimmed mean.** The goal is to estimate the (shifted) distance to optimum, $\|w_t - w^*\|_\Sigma^2 + \sigma^2$, up to some constant multiplicative error. Note that this is precisely the task of estimating the variance of the residual $b_i = y_i - w_t^\top x_i$. When there is no adversarial corruptions and no privacy constraints, we can simply use the empirical variance estimator $(1/n)\sum_{i\in[n]}(y_i - w_t^\top x_i)^2$ to obtain a good estimate. However, the empirical variance estimator is not robust against adversarial corruptions since one outlier can make the estimate arbitrarily large. A classical idea is using the *trimmed estimator* from (Tukey & McLaughlin, 1963), which throws away the $2\alpha$ fraction of residuals $b_i$ with the largest magnitude. For datasets with resilience property as assumed in this paper, this will guarantee an accurate estimate of the distance to optimum in the presence of $\alpha$ fraction of corruptions.

To make the estimator private, it is tempting to simply add a Laplacian noise to the estimate. However, the sensitivity of the trimmed estimator is unknown and depends on the distance to the optimum

that we aim to estimate; we cannot determine the variance of the Laplacian noise we need to generate. Instead, we propose to partition the dataset into $k$ batches, compute an estimate for each batch, and form a histogram with over those $k$ estimates. Using a private histogram mechanism with geometrically increasing bin sizes, we propose using the bin with the most estimates to guarantee a constant factor approximation of the distance to the optimum. We describe the algorithm as follows.

---

**Algorithm 2:** Robust Private Distance Estimator

**Input:** $S_2 = \{(x_i, y_i)\}_{i=1}^n$, current weight $w_t$, privacy $(\varepsilon_0, \delta_0)$, $\bar{\alpha}, \zeta$

1  Let $b_i \leftarrow (y_i - w_t^\top x_i)^2$, for all $i \in [n]$ and $\tilde{S} \leftarrow \{b_i\}_{i=1}^n$.

2  Partition $\tilde{S}$ into $k = \lfloor C_1 \log(1/(\delta_0\zeta))/\varepsilon_0 \rfloor$ subsets of equal size and let $G_j$ be the $j$-th partition.

3  For $j \in [k]$, denote $\psi_j$ as the $(1 - 3\bar{\alpha})$-quantile of $G_j$ and $\phi_j \leftarrow \frac{1}{|G_j|} \sum_{i \in G_j} b_i \mathbf{1}\{b_i \leq \psi_j\}$.

4  Partition $[0, \infty)$ into bins of geometrically increasing intervals
   $\Omega := \left\{\ldots, \left[2^{-1}, 1\right), [1, 2), \left[2, 2^2\right), \left[2^2, 2^3\right), \ldots\right\} \cup \{[0, 0]\}$

5  Run $(\varepsilon_0, \delta_0)$-DP histogram learner of Lemma B.1 on $\{\phi_j\}_{j=1}^k$ over $\Omega$

6  **if** all the bins are empty **then** Return $\bot$

7  Let $[\ell, r]$ be a non-empty bin that contains the maximum number of points in the DP histogram

8  **return** $\ell$

---

This algorithm gives an estimate of the distance up to a constant multiplicative error as we show in the following theorem. We provide a proof in App. D.

**Theorem 4.** *Algorithm 2 is $(\varepsilon_0, \delta_0)$-DP. For an $\alpha_{\mathrm{corrupt}}$-corrupted dataset $S_2$ and an upper bound $\bar{\alpha}$ on $\alpha_{\mathrm{corrupt}}$ that satisfy Assumption 1 and $37C_2K^2 \cdot \bar{\alpha} \log^{2a}(1/(6\bar{\alpha})) \leq 1/4$ and any $\zeta \in (0, 1)$, if*

$$n = O\left(\frac{(d + \log((\log(1/(\delta_0\zeta)))/\varepsilon_0\zeta))(\log(1/(\delta_0\zeta)))}{\bar{\alpha}^2\varepsilon_0}\right), \tag{6}$$

*with a large enough constant then, with probability $1 - \zeta$, Algorithm 2 returns $\ell$ such that $\frac{1}{4}(\|w_t - w^*\|_\Sigma^2 + \sigma^2) \leq \ell \leq 4(\|w_t - w^*\|_\Sigma^2 + \sigma^2)$.*

**Remark 4.1.** *While DP-STAT (Algorithm 3 in Varshney et al. (2022)) can also be used to estimate $\|w_t - w^*\|_\Sigma + \sigma$ (and it would not change the ultimate sample complexity in its dependence on $\kappa$, $d$, $\varepsilon$, and $n$), there are three important improvements we make: $(i)$ DP-STAT requires the knowledge of $\|w^*\|_\Sigma + \sigma$; $(ii)$ our utility guarantee has improved dependence in $K$ and $\log^{2a}(n)$; and $(iii)$ Algorithm 2 is robust against label corruption.*

**Upper bound on clipped good data points.** Using the above estimated distance to the optimum in selecting a threshold $\theta_t$, we also need to ensure that we do not clip too many clean data points. The tolerance in our algorithm to reach the desired level of accuracy is clipping $O(\alpha)$ fraction of clean data points. This is ensured by the following lemma, and we provide a proof in Appendix E.

**Lemma 4.2.** *Under Assumptions 1, if $\theta_t \geq \sqrt{9C_2K^2 \log^{2a}(1/(2\alpha))} \cdot (\|w^* - w_t\|_\Sigma + \sigma)$, then $\left|\{i \in S_3 \cap S_{\mathrm{good}} : \left|w_t^\top x_i - y_i\right| \geq \theta_t\}\right| \leq \alpha n$, for all $t \in [T]$.*

## 5 EXPERIMENTAL RESULTS

### 5.1 DP LINEAR REGRESSION

We present experimental results comparing our proposed technique (DP-ROBGD) with other baselines. We consider non-corrupted regression in this section and defer corrupted regression to the next section. We begin by describing the problem setup and the baseline algorithms first.

**Experiment Setup.** We generate data for all the experiments using the following generative model. The parameter vector $(w^*)$ is uniformly sampled from the surface of a unit sphere. The covariates $\{x_i\}_{i=1}^n$ are first sampled from $\mathcal{N}(0, \Sigma)$ and then projected to unit sphere. We consider diagonal covariances $\Sigma$ of the following form: $\Sigma[0, 0] = \kappa$, and $\Sigma[i, i] = 1$ for all $i \geq 1$. Here $\kappa \geq 1$ is the condition number of $\Sigma$. We generate noise $z_i$ from uniform distribution over $[-\sigma, \sigma]$. Finally,

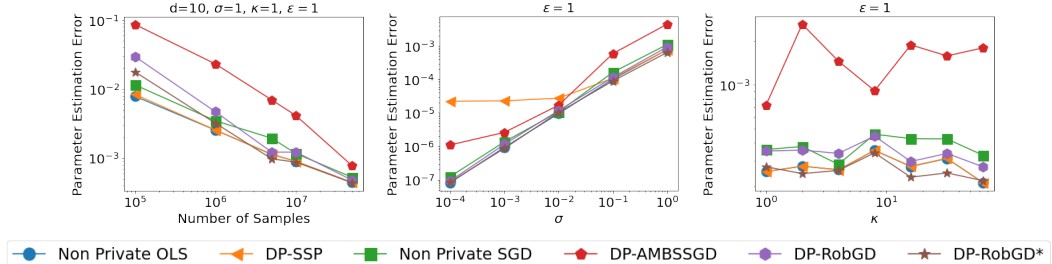

Figure 1: Performance of various techniques on DP linear regression. $d = 10$ in all the experiments. $n = 10^7, \kappa = 1$ in the $2^{nd}$ experiment. $n = 10^7, \sigma = 1$ in the $3^{rd}$ experiment.

the response variables are generated as follows $y_i = \langle x_i, w^* \rangle + z_i$. All the experiments presented below are repeated 5 times and the averaged results are presented. We set the DP parameters $(\epsilon, \delta)$ as $\epsilon = 1, \delta = \min(10^{-6}, n^{-2})$. Experiments for $\epsilon = 0.1$ can be found in the Appendix.

**Baseline Algorithms.** We compare our estimator with the following baseline algorithms:

- *Non private algorithms:* ordinary least squares (OLS), one-pass stochastic gradient descent with tail-averaging (SGD). For SGD, we use a constant step-size of $1/(2\lambda_{max})$ with $n/T$ minibatch size, where $T = 3\kappa \log n$.
- *Private algorithms:* sufficient statistics perturbation (DP-SSP) (Foulds et al., 2016; Wang, 2018), differentially private stochastic gradient descent (DP-AMBSSGD) (Varshney et al., 2022). DP-SSP had the best empirical performance among numerous techniques studied by Wang (2018), and DP-AMBSSGD has the best known theoretical guarantees. The DP-SSP algorithm involves releasing $X^T X$ and $X^T \mathbf{y}$ differentially privately and computing $\widehat{(X^T X)}^{-1} \widehat{X^T \mathbf{y}}$. DP-AMBSSGD is a private version of SGD where the DP noise is set adaptively according to the excess error in each iteration. For both the algorithms, we use the hyper-parameters recommended in their respective papers. To improve the performance of DP-AMBSSGD, we reduce the clipping threshold recommended by the theory by a constant factor.

**DP-ROBGD.** We implement Algorithm 1 with the following key changes. Instead of relying on PrivateNormEstimator to estimate $\Gamma$, we set it to its true value $\text{Tr}(\Sigma)$. This is done for a fair comparison with DP-AMBSSGD which assumes the knowledge of $\text{Tr}(\Sigma)$. Next, we use 20% of the samples to compute $\gamma_t$ in line 5 (instead of the 50% stated in Algorithm 1). In our experiments we also present results for a variant of our algorithm called DP-ROBGD* which outputs the best iterate based on $\gamma_t$, instead of the last iterate. One could also perform tail-averaging instead of picking the best iterate. Both these modifications are primarily used to reduce the variance in the output of Algorithm 1 and achieved similar performance in our experiments.

**Results.** Figure 1 presents the performance of various algorithms as we vary $n, \kappa, \sigma$. It can be seen that DP-ROBGD outperforms DP-AMBSSGD in almost all the settings. DP-SSP has poor performance when the noise $\sigma$ is low, but performs slightly better than DP-ROBGD in other settings. A major drawback of DP-SSP is its computational complexity which scales as $O(nd^2 + d^\omega)$. In contrast, the computational complexity of DP-ROBGD has smaller dependence on $d$ and scales as $\tilde{O}(nd\kappa)$. Thus the latter is more computationally efficient for high-dimensional problems.

## 5.2 DP ROBUST LINEAR REGRESSION

We now illustrate the robustness of our algorithm. We consider the same experimental setup as above and randomly corrupt $\alpha$ fraction of the response variables by setting them to 1000. The figure on the right presents the results from this experiment. It can be seen that none of the baselines are robust to adversarial corruptions. They can be made arbitrarily bad by in-

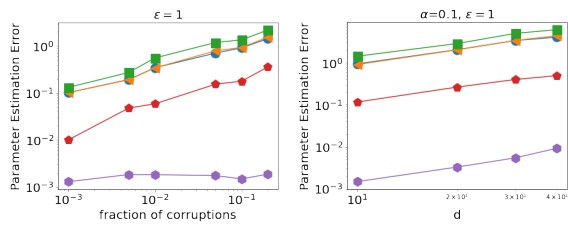

creasing the magnitude of corruptions. In contrast, DP-ROBGD is able to handle the corruptions well. More experimental results on a harder adversary can be found in the Appendix.

## 6 SKETCH OF THE MAIN IDEAS IN THE ANALYSIS

We provide the main ideas behind the proof of Theorem 3. The privacy proof is straightforward since no matter what clipping threshold we get from private norm estimator and private distance estimator, the noise we add is always proportionally to the clipping threshold which guaranteed privacy. The focus of this section will be on the utility of the algorithm.

The proof of the utility heavily relies on the *resilience* Steinhardt et al. (2017) (also known as *stability* Diakonikolas & Kane (2019)), which states that given a large enough sample set $S$, varies statistics (for example, sample mean and sample variance) of any large enough subset of $S$ will be close to each other. We provide the formal definition of resilience in Appendix C.

The main effort for proving Theorem 3 lies in the analysis of the gradient descent algorithm. Without clipping and added noise for differential privacy, convergence property of gradient descent for linear regression is well known. The convergence proof of noisy gradient descent is also relatively straightforward. However, our algorithm requires clipping and added noise together for robustness and privacy purposes, and the key difference between our setting and the classical setting is the existence of adversarial bias and random noise in the gradient. We give an overview of the proof of our robust and private gradient descent as follows.

First we introduce some notations. Let $g_i = (x_i^\top w_t - y_i)x_i$ be the raw gradient, $\tilde{g}_i = \mathrm{clip}_{\theta_t}(x_i^\top w_t - y_i)\mathrm{clip}_{\Theta}(x_i)$ be the clipped gradient. Note that when the data follows from our distributional assumption, $\mathrm{clip}_{\Theta}(x_i) = x_i$ for $i \in S_{\mathrm{good}}$. We can write down one step of gradient update as follows:

$$w_{t+1} - w^* = w_t - \eta \left( \frac{1}{n} \sum_{i \in S} \tilde{g}_i^{(t)} + \phi_t \nu_t \right) - w^*$$

$$= \left( \mathbf{I} - \frac{\eta}{n} \sum_{i \in G} x_i x_i^\top \right)(w_t - w^*) + \frac{\eta}{n} \sum_{i \in G} x_i z_i + \frac{\eta}{n} \sum_{i \in G} (g_i^{(t-1)} - \tilde{g}_i^{(t-1)}) - \frac{\eta}{n} \sum_{i \in S_{\mathrm{bad}}} \tilde{g}_i^{(t)} - \eta \phi_t \nu_t$$

In the above equation, the first term is a contraction, meaning $w_t$ is moving toward $w^*$. The second term captures the noise from the randomness of the data set. The third term captures the bias introduced by the clipping operation, the fourth term $\frac{\eta}{n} \sum_{i \in S_{\mathrm{bad}}} \tilde{g}_i^{(t)}$ captures the bias introduced by the adversarial datapoints, the fifth term captures the added Gaussian noise. The second term is standard and relatively easy to control, and our main focus is on the last three terms.

The third term $\frac{\eta}{n} \sum_{i \in G} (g_i^{(t-1)} - \tilde{g}_i^{(t-1)})$ can be controlled using the resilience property. We prove that with our estimated threshold, the clipping will only affect a small amount of datapoints, whose contribution to the gradient is small collectively. The fourth term $\frac{\eta}{n} \sum_{i \in S_{\mathrm{bad}}} \tilde{g}_i^{(t)} = \frac{\eta}{n} \sum_{i \in S_{\mathrm{bad}}} \mathrm{clip}_{\theta_t}(x_i^\top w_t - y_i)x_i$ can be controlled since there is only a small amount data points whose label is corrupted, the $\mathrm{clip}_{\theta_t}(x_i^\top w_t - y_i)$ is controlled by the clipping threshold and the $x_i$ part satisfies resilience property which implies a small, say $S_{\mathrm{bad}}$, must have small $\|\sum_{i \in S_{\mathrm{bad}}} x_i\|$.

Now we have controlled the deterministic bias. Then, we upper bound the fifth term, which is the noise introduced by the Gaussian noise for the purpose of privacy, and show the expected prediction error decrease in every gradient step. The difficulty is that, since our clipping threshold is adaptive, the decrease of the estimation error depends on the estimation error of all the previous steps. This causes that in some iterations, the estimation error actually increase. In order to get around this, we split the iterations into length $\kappa$ chunks, and argue that the maximum estimation error in a chunk must be a constant factor smaller than the previous chunk. This implies we will reach the desired error with in $\tilde{O}(\kappa)$ steps.

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

APPENDIX

## A  RELATED WORK

**Differentially private optimization.** There is a long line of work at the intersection of differentially privacy and optimization (Chaudhuri et al., 2011; Kifer et al., 2012; Bassily et al., 2014; Song et al., 2013; Bassily et al., 2019; Wu et al., 2017; Andrew et al., 2021; Feldman et al., 2020; Song et al., 2020; Asi et al., 2021; Kulkarni et al., 2021; Kamath et al., 2021; Zhang et al., 2022). As one of the most well-studied problem in differentially privacy, DP Empirical Risk Minimization (DP-ERM) aims to minimize the empirical risk $(1/n) \sum_{i \in S} \ell(x_i; w)$ privately. The optimal excess empirical risk for approximate DP (i.e., $\delta > 0$) is known to be $GD \cdot \sqrt{d}/(\varepsilon n)$, where the loss $\ell$ is convex and $G$-Lipschitz with respect to the data, and $D$ is the diameter of the convex parameter space (Bassily et al., 2014). This bound can be achieved by several DP-SGD methods, e.g., (Song et al., 2013; Bassily et al., 2014), with different computational complexities. Differentially private stochastic convex optimization considers minimizing the population risk $\mathbb{E}_{x \sim \mathcal{D}}[\ell(x, w)]$, where data is drawn i.i.d. from some unknown distribution $\mathcal{D}$. Using some variations of DP-SGD, Bassily et al. (2019) and Feldman et al. (2020) achieves a population risk of $GD(1/\sqrt{n} + \sqrt{d}/(\varepsilon n))$.

**DP linear regression.** Applying above results for the linear model, by observing that $G = O(d)$ if $D = O(1)$, the sample complexity required for achieving generalization error is $n = d^2$. Existing works for DP linear regression, for example (Vu & Slavkovic, 2009; Kifer et al., 2012; Mir, 2013; Dimitrakakis et al., 2014; Wang et al., 2015; Foulds et al., 2016; Minami et al., 2016; Wang, 2018; Sheffet, 2019; Wang & Gu, 2019; Hu et al., 2022) typically consider deterministic data. Under the i.i.d. Gaussian data setting, this translates into a sample complexity of $n = d^{3/2}/(\varepsilon\alpha)$, where the extra $d^{1/2}$ due to the fact that no statistical assumptions are made. For i.i.d. sub-Weibull data, recent work (Varshney et al., 2022) achieved nearly optimal excess population risk $d/n + d^2/(\varepsilon^2 n^2)$ using DP-SGD with adaptive clipping, up to extra factors on the condition number. This is closest to our work and we provide detailed comparisons in Sections 2.2 and 3.2. Under Gaussian assumptions, Milionis et al. (2022) analyze linear regression algorithm with sub-optimal guarantees. (Dwork & Lei, 2009; Alabi et al., 2020; Amin et al., 2022; Liu et al., 2022b) also consider using robust statistics like Tukey median (Tukey, 1975) or Theil–Sen estimator (Theil, 1950) for differentially private regression. However, (Dwork & Lei, 2009; Amin et al., 2022) lack utility guarantees and (Alabi et al., 2020) is restricted to one-dimensional data. Liu et al. (2022b) achieves optimal sample complexity but takes exponential time.

**Robust linear regression.** Robust mean estimation and linear regression have been studied for a long time in the statistics community (Tukey & McLaughlin, 1963; Huber, 1992; Tukey, 1975). However, for high dimensional data, these estimators generalizing the notion of median to higher dimensions are typically computationally intractable. Recent advances in the filter-based algorithms, e.g., (Diakonikolas et al., 2017; 2020; 2019a; 2018; Cheng et al., 2019; Dong et al., 2019), achieve nearly optimal guarantees for mean estimation in time linear in the dimension of the dataset. Motivated by the filter algorithms, Diakonikolas et al. (2019c;b); Prasad et al. (2018); Pensia et al. (2020); Cherapanamjeri et al. (2020); Jambulapati et al. (2020) achieved nearly optimal rate with $d$ samples for robust linear regression, where both data $x_i$ and label $y_i$ are corrupted. Another type of efficient methods that achieve similar rates and sample complexity in polynomial time is based on sum-of-square proofs (Klivans et al., 2018; Bakshi & Prasad, 2021), which can be computationally expensive in practice. Zhu et al. (2019); Brown et al. (2021); Liu et al. (2022b) achieve nearly optimal rates using $d$ samples but require exponential time complexities. An important special case of adversarial corruption is when the adversary only corrupts the response variable in supervised learning (Khetan et al., 2018) and also in unsupervised learning (Thekumparampil et al., 2018). For linear regression, when there is only label corruptions, (Bhatia et al., 2015; Dalalyan & Thompson, 2019; Kong et al., 2022) achieve nearly optimal rates with $O(d)$ samples. Under the oblivious label corruption model, i.e., the adversary only corrupts a fraction of labels in complete ignorance of the data, (Bhatia et al., 2017; Suggala et al., 2019) provide consistent estimator $\hat{w}_n$ such that $\lim_{n \to \infty} \mathbb{E}[\hat{w}_n - w^*]_2 = 0$ with $O(d)$ samples.

**Robust and private linear regression.** Under the settings of both DP and data corruptions, the only algorithm by Liu et al. (2022b) achieves nearly optimal rates $\alpha \log(1/\alpha)\sigma$ with optimal sample complexities of $d/\alpha^2 + d/(\varepsilon\alpha)$. However, their algorithm requires exponential time complexities.

## B  PRELIMINARY ON DIFFERENTIAL PRIVACY

Our algorithm builds upon two DP primitive: Gaussian mechanism and private histogram. The Gaussian mechanism is one examples of a larger family of mechanisms known as output perturbation mechanisms. In practice, it is possible to get better utility trade-off for a output perturbation mechanism by carefully designing the noise, such as the stair-case mechanism which are shown to achieve optimal utility in the variance (Geng et al., 2015) and also in hypothesis testing (Kairouz et al., 2014). However, the gain is only by constant factors, which we do not try to optimize in this paper. We provide a reference for the private histogram below.

**Lemma B.1** (Stability-based histogram (Karwa & Vadhan, 2017, Lemma 2.3)). *For every $K \in \mathbb{N} \cup \infty$, domain $\Omega$, for every collection of disjoint bins $B_1, \dots, B_K$ defined on $\Omega$, $n \in \mathbb{N}$, $\varepsilon \geq 0, \delta \in (0, 1/n)$, $\beta > 0$ and $\alpha \in (0, 1)$ there exists an $(\varepsilon, \delta)$-differentially private algorithm $M : \Omega^n \to \mathbb{R}^K$ such that for any set of data $X_1, \dots, X_n \in \Omega^n$*

*1. $\hat{p}_k = \frac{1}{n} \sum_{X_i \in B_k} 1$*

*2. $(\tilde{p}_1, \dots, \tilde{p}_K) \leftarrow M(X_1, \dots, X_n)$, and*

*3.*

$$n \geq \min \left\{ \frac{8}{\varepsilon \beta} \log(2K/\alpha), \frac{8}{\varepsilon \beta} \log(4/\alpha\delta) \right\}$$

*then,*

$$\mathbb{P}(|\tilde{p}_k - \hat{p}_k| \leq \beta) \geq 1 - \alpha$$

When the databse is accessed multiple times, we use the following composition theorems to account for the end-to-end privacy leakage.

**Lemma B.2** (Parallel composition McSherry (2009)). *Consider a sequence of interactive queries $\{q_k\}_{k=1}^K$ each operating on a subset $S_k$ of the database and each satisfying $(\varepsilon, \delta)$-DP. If $S_k$'s are disjoint then the composition $(q_1(S_1), q_2(S_2), \dots, q_K(S_K))$ is $(\varepsilon, \delta)$-DP.*

**Lemma B.3** (Serial composition Dwork & Roth (2014)). *If a database is accessed with an $(\varepsilon_1, \delta_1)$-DP mechanism and then with an $(\varepsilon_2, \delta_2)$-DP mechanism, then the end-to-end privacy guarantee is $(\varepsilon_1 + \varepsilon_2, \delta_1 + \delta_2)$-DP.*

In most modern privacy analysis of iterative processes, advanced composition theorem from Kairouz et al. (2015) gives tight accountant for the end-to-end privacy budget. It can be improved for specific mechanisms using tighter accountants, e.g., in Mironov (2017); Girgis et al. (2021); Wang et al. (2019); Zhu et al. (2022); Gopi et al. (2021).

**Lemma B.4** (Advanced composition Kairouz et al. (2015)). *For $\varepsilon \leq 0.9$, an end-to-end guarantee of $(\varepsilon, \delta)$-differential privacy is satisfied if a database is accessed $k$ times, each with a $(\varepsilon/(2\sqrt{2k \log(2/\delta)}), \delta/(2k))$-differential private mechanism.*

## C  DEFINITION OF RESILIENCE

**Definition C.1** ((Liu et al., 2022b, Definition 23)). *For some $\alpha \in (0, 1)$, $\rho_1 \in \mathbb{R}_+$, $\rho_2 \in \mathbb{R}_+$, and $\rho_3 \in \mathbb{R}_+$, $\rho_4 \in \mathbb{R}_+$, we say dataset $S_{\text{good}} = \{(x_i \in \mathbb{R}^d, y_i \in \mathbb{R})\}_{i=1}^n$ is $(\alpha, \rho_1, \rho_2, \rho_3, \rho_4)$-resilient with respect to $(w^*, \Sigma, \sigma)$ for some $w^* \in \mathbb{R}^d$, positive definite $\Sigma \succ 0 \in \mathbb{R}^{d \times d}$, and $\sigma > 0$ if for any*

$T \subset S_{\text{good}}$ *of size* $|T| \geq (1 - \alpha)n$, *the following holds for all* $v \in \mathbb{R}^d$:

$$\left| \frac{1}{|T|} \sum_{(x_i, y_i) \in T} \langle v, x_i \rangle (y_i - x_i^\top w^*) \right| \leq \rho_1 \sqrt{v^\top \Sigma v} \, \sigma \ , \tag{7}$$

$$\left| \frac{1}{|T|} \sum_{x_i \in T} \langle v, x_i \rangle^2 - v^\top \Sigma v \right| \leq \rho_2 v^\top \Sigma v \ , \tag{8}$$

$$\left| \frac{1}{|T|} \sum_{(x_i, y_i) \in T} (y_i - x_i^\top w^*)^2 - \sigma^2 \right| \leq \rho_3 \sigma^2 \ , \tag{9}$$

$$\left| \frac{1}{|T|} \sum_{(x_i, y_i) \in T} \langle v, x_i \rangle \right| \leq \rho_4 \sqrt{v^\top \Sigma v} \ . \tag{10}$$

## D  PROOF OF THEOREM 4 ON THE PRIVATE DISTANCE ESTIMATION

We first analyze the privacy. Changing a data point $(x_i, y_i)$ can affect at most one partition in $\{G_j\}_{j=1}^k$. This would affect at most two histogram bins, increasing the count of one bin by one and decreasing the count in another bin by one. Under such a bounded $\ell_1$ sensitivity, the privacy guarantees follows from Lemma B.1.

Next, we analyze the utility. In the (private) histogram step, we claim that at most only two consecutive bins can be occupied by any $\phi_j$'s. This is also true for the private histogram, because the private histogram of Lemma B.1 adds noise to non-empty bins only. By Lemma B.1, if $k \geq c \log(1/(\delta_0 \zeta_0))/\varepsilon_0$, one of these two intervals (the union of which contains the true distance $\|w_t - w^*\|_\Sigma^2 + \sigma^2$) is released. This results in a multiplicative error bound of four, as the bin size increments by a factor of two.

To show that only two bins are occupied, we show that all $\phi_j$'s are close to the true distance. We first show that each partition contains at most $2\alpha_{\text{corrupt}}$ fraction of corrupted samples and thus all partitions are $(2\bar{\alpha}, 6\bar{\alpha}, 6\hat{\rho}, 6\hat{\rho}, 6\hat{\rho}')$-corrupt good, where $\hat{\rho}(C_2, K, a, \bar{\alpha}) = C_2 K^2 \bar{\alpha} \log^{2a}(1/6\bar{\alpha})$ and $\hat{\rho}'(C_2, K, a, \bar{\alpha}) = C_2 K \bar{\alpha} \log^a(1/6\bar{\alpha})$, as defined in Definition J.6.

Let $B = \lfloor n/k \rfloor$ be the sample size in each partition. Let $\zeta_0 = \zeta/2$. Since the partition is drawn uniformly at random, for each partition $G_j$, the number of corrupted samples $\alpha'n$ satisfies $\alpha'n \sim$ Hypergeometric$(n, \alpha_{\text{corrupt}}n, n/k)$. The tail bound gives that with probability $1 - \zeta_0$,

$$\alpha' \leq \alpha_{\text{corrupt}} + (k/n) \log(2/\zeta_0) \leq 2\bar{\alpha} \ ,$$

where the last inequality follows from the fact that the corruption level is bounded by $\alpha_{\text{corruption}} \leq \bar{\alpha}$ and the assumption on the sample size in Eq. (6) which implies $n \gtrsim \log(1/(\delta_0 \zeta_0)) \log(1/\zeta_0)/(\bar{\alpha}\varepsilon_0)$.

For a particular subset $G_j$, Lemma J.7 implies that if $B = O((d + \log(1/\zeta_0))/\bar{\alpha}^2)$, then $G_j$ is $(\alpha', 6\bar{\alpha}, 6\hat{\rho}, 6\hat{\rho}, 6\hat{\rho}')$-corrupt good set with respect to $(w^*, \Sigma, \sigma)$ from Assumption 1. This means that there exists a constant $C_2 > 0$ such that for any $T_1 \subset S_{\text{good}}$ with $|T_1| \geq (1 - 6\bar{\alpha})B$, we have

$$\left| \frac{1}{|T_1|} \sum_{i \in T_1} \langle x_i, w^* - w_t \rangle^2 - \|w^* - w_t\|_\Sigma^2 \right| \leq 6 C_2 K^2 \bar{\alpha} \log^{2a}(1/(6\bar{\alpha})) \|w^* - w_t\|_\Sigma^2 \ ,$$

$$\left| \frac{1}{|T_1|} \sum_{i \in T_1} z_i^2 - \sigma^2 \right| \leq 6 C_2 K^2 \bar{\alpha} \log^{2a}(1/(6\bar{\alpha})) \sigma^2 \ ,$$

and

$$\left| \frac{1}{|T_1|} \sum_{i \in T_1} z_i \langle x_i, w^* - w_t \rangle \right| \leq 6 C_2 K^2 \bar{\alpha} \log^{2a}(1/(6\bar{\alpha})) \|w^* - w_t\|_\Sigma \sigma \ .$$

Note that for $i \in S_{\text{good}}$, $b_i = z_i^2 + 2z_i(w^* - w_t)^\top x_i + (w^* - w_t)^\top x_i x_i^\top (w^* - w_t)$. By the triangular inequality, we know, under above conditions,

$$\left| \frac{1}{|T_1|} \sum_{i \in T_1} b_i - \|w^* - w_t\|_\Sigma^2 - \sigma^2 \right| \leq 12 C_2 K^2 \bar{\alpha} \log^{2a}(1/(6\bar{\alpha}))(\|w^* - w_t\|_\Sigma^2 + \sigma^2) \ . \tag{11}$$

Which also implies that any subset $T_2 \subset S_{\text{good}}$ and $|T_2| \leq 6\bar{\alpha}|S_{\text{good}}|$, we have

$$\left| \frac{1}{|T_2|} \sum_{i \in T_2} b_i - \|w^* - w_t\|_{\Sigma}^2 - \sigma^2 \right| \leq 12C_2K^2 \log^{2a}(1/(6\bar{\alpha}))(\|w^* - w_t\|_{\Sigma}^2 + \sigma^2) . \tag{12}$$

Recall that $\psi_j$ is the $(1-3\bar{\alpha})$-quantile of the dataset $G_j$. Let $T := \{i \in S_{\text{good}} : b_i \leq \psi_j\}$, where with a slight abuse of notations, we use $S_{\text{good}}$ to denote the set of uncorrupted samples corresponding to $G_j$ and $S_{\text{bad}}$ to denote the set of corrupted samples corresponding to $G_j$. Since the corruption is less than $\alpha'$, we know $(1 - 3\bar{\alpha} - \alpha')B \leq |T| \leq (1 - 3\bar{\alpha} + \alpha')B$. By our assumption that $\alpha' \leq 2\bar{\alpha}$, we have $|\bar{E}| \geq (3\bar{\alpha} - \alpha')B \geq \bar{\alpha}B$ where $\bar{E} := S_{\text{good}} \setminus E$. Using Eq. (12) with a choice of $T_2 = \bar{E}$, we get that

$$\min_{i \in \bar{E}} b_i - \|w^* - w_t\|_{\Sigma}^2 - \sigma^2 \leq 12C_2K^2 \log^{2a}(1/(6\bar{\alpha}))(\|w^* - w_t\|_{\Sigma}^2 + \sigma^2) . \tag{13}$$

This implies that

$$\psi_j \leq 12C_2K^2 \log^{2a}(1/(6\bar{\alpha}))(\|w^* - w_t\|_{\Sigma}^2 + \sigma^2). \tag{14}$$

Hence

$$\left| \phi_j - \|w^* - w_t\|_{\Sigma}^2 - \sigma^2 \right| = \left| \frac{1}{B} \sum_{i \in G_j} b_i \cdot \mathbf{1}\{b_i \leq \psi_j\} - \|w^* - w_t\|_{\Sigma}^2 - \sigma^2 \right|$$

$$= \left| \frac{1}{B} \sum_{i \in T} b_i - \|w^* - w_t\|_{\Sigma}^2 - \sigma^2 \right| + \left| \frac{1}{B} \sum_{i \in S_{\text{bad}}} b_i \cdot \mathbf{1}\{b_i \leq \psi_j\} \right|$$

$$\leq 37C_2K^2 \cdot \bar{\alpha} \log^{2a}(1/(6\bar{\alpha}))(\|w^* - w_t\|_{\Sigma}^2 + \sigma^2), \tag{15}$$

where we applied Eq. (14) and Eq. (11) in the last inequality.

On a fixed partition $G_j$, we showed that if $B = O((d + \log(1/\zeta_0))/\bar{\alpha}^2)$ then, with probability $1 - \zeta_0$, $|\phi_j - \|w^* - w_t\|_{\Sigma}^2 - \sigma^2| \leq \frac{1}{4}(\|w^* - w_t\|_{\Sigma}^2 + \sigma^2)$, which follows from our assumption that $37C_2K^2 \cdot \bar{\alpha} \log^{2a}(1/(6\bar{\alpha})) \leq 1/4$. Using an union bound for all subsets, we know if $B = O((d + \log(k/\zeta_0))/\bar{\alpha}^2)$, then $1 - \zeta_0$, $|\phi_j - \|w^* - w_t\|_{\Sigma}^2 - \sigma^2| \leq \frac{1}{4}(\|w^* - w_t\|_{\Sigma}^2 + \sigma^2)$ holds for all $j \in [k]$. Since the upper bound lower bound ratio is $5/3$ which is less than 2. All the $\phi_j$ must lie in two bins, which will result in a factor of 4 multiplicative error.

## E  PROOF OF LEMMA 4.2 ON THE UPPER BOUND ON CLIPPED GOOD POINTS

Let $\hat{\rho}(C_2, K, a, \alpha) = 2C_2K^2\alpha \log^{2a}(1/(2\alpha))$ and $\hat{\rho}'(C_2, K, a, \alpha) = 2C_2K\alpha \log^a(1/(2\alpha))$. Lemma J.7 implies that if $n = O((d + \log(1/\zeta))/(\alpha^2))$ with a large enough constant, then there exists a universal constant $C_2$ such that $S_3$ is, with respect to $(w^*, \Sigma, \sigma)$, $(\alpha_{\text{corrupt}}, 2\alpha, \hat{\rho}, \hat{\rho}, \hat{\rho}, \hat{\rho}')$-corrupt good. The rest of the proof is under this (deterministic) resilience condition. By the resilience property in Eq. (8), we know for any $T \subset S_{\text{good}}$ with $|T| \geq (1 - 2\alpha)n$,

$$\left| \frac{1}{|T|} \sum_{i \in T} (w^* - w_t)^\top x_i x_i^\top (w^* - w_t) - \|w^* - w_t\|_{\Sigma}^2 \right| \leq 2C_2K^2\alpha \log^{2a}(1/(2\alpha))\|w^* - w_t\|_{\Sigma}^2 . \tag{16}$$

Let $E := \{i \in S_{\text{good}} : (w^* - w_t)^\top x_i x_i^\top (w^* - w_t) > \|w^* - w_t\|_{\Sigma}^2(8C_2K^2 \log^{2a}(1/(2\alpha)) + 1)\}$. Denote $\tilde{\alpha} := |E|/n$. We want to show that $\tilde{\alpha} \leq \alpha/2$. Let $T$ be the set of points that contain the smallest $1 - \alpha/2$ fraction in $\{(w^* - w_t)^\top x_i x_i^\top (w^* - w_t)\}_{i \in S_{\text{good}}}$. We know $|T| = (1 - \alpha/2)n \geq (1 - 2\alpha)n$. To prove by contradiction, suppose $\tilde{\alpha} > \alpha/2$, which means all data points in $S_{\text{good}} \setminus T$ are larger than $\|w^* - w_t\|_{\Sigma}^2(8C_2K^2 \log^{2a}(1/(2\alpha)) + 1)$. From resilience property in Eq. (16), we

know

$$\frac{1}{n} \sum_{i \in S_{\text{good}}} (w^* - w_t)^\top x_i x_i^\top (w^* - w_t)$$

$$= \frac{1}{n} \sum_{i \in T} (w^* - w_t)^\top x_i x_i^\top (w^* - w_t) + \frac{1}{n} \sum_{i \in S_{\text{good}} \setminus T} (w^* - w_t)^\top x_i x_i^\top (w^* - w_t)$$

$$\geq \left(1 - \frac{\alpha}{2}\right) \left(1 - 2C_2 K^2 \alpha \log^{2a}(\frac{1}{2\alpha})\right) \|w^* - w_t\|_\Sigma^2 + \frac{\alpha}{2} (8C_2 K^2 \log^{2a}(\frac{1}{2\alpha}) + 1) \|w^* - w_t\|_\Sigma^2$$

$$> (1 + 2C_2 K^2 \alpha \log^{2a}(1/2\alpha)) \|w^* - w_t\|_\Sigma^2 \,,$$

which contradicts Eq. (16) for $S_{\text{good}}$. This shows $\tilde{\alpha} \leq \alpha/2$.

Similarly, we can show that $\left|\left\{i \in S_{\text{good}} : z_t^2 > \sigma^2 (8C_2 K^2 \log^{2a}(1/(2\alpha)) + 1)\right\}\right| \leq \alpha/2$. This means the rest $(1 - \alpha)n$ points in $S_{\text{good}}$ satisfies $\sqrt{(w^* - w_t)^\top x_i x_i^\top (w^* - w_t)} + |z_i| \leq (\|w_t - w^*\| + \sigma)\sqrt{(8C_2 K^2 \log^{2a}(1/(2\alpha)) + 1)}$. Note that for all $i \in S_{\text{good}}$, we have

$$\begin{aligned} |x_i^\top w_t - y_i| = \left|x_i^\top (w_t - w^*) - z_i\right| \\ \leq |x_i^\top (w_t - w^*)| + |z_i| \\ \leq \left(\sqrt{(w^* - w_t)^\top x_i x_i^\top (w^* - w_t)} + |z_i|\right) \,. \end{aligned}$$

By our assumption that $C_2 K^2 \log^{2a}(1/(2\bar{\alpha})) \geq 1$ which follows from Assumption 2, we have

$$\left|\left\{i \in S_{\text{good}} : \|x_i^\top w_t - y_i\| \leq (\|w_t - w^*\| + \sigma)\sqrt{9 C_2 K^2 \log^{2a}(1/(2\alpha))}\right\}\right| \geq (1 - \alpha)n \,. \quad (17)$$

## F   PRIVATE NORM ESTIMATION: ALGORITHM AND ANALYSIS

---

**Algorithm 3:** Private Norm Estimator

**Input:** $S_1 = \{(x_i, y_i)\}_{i=1}^n$, target privacy $(\varepsilon_0, \delta_0)$, failure probability $\zeta$.

1 Let $a_i \leftarrow \|x_i\|^2$. Let $\tilde{S} = \{a_i\}_{i=1}^n$.
2 Partition $\tilde{S}$ into $k = \lfloor C_1 \log(1/(\delta_0 \zeta))/\varepsilon \rfloor$ subsets of equal size and let $G_j$ be the $j$-th partition.
3 For each $j \in [k]$, denote $\psi_j = (1/|G_j|) \sum_{i \in G_j} a_i$.
4 Partition $[0, \infty)$ into bins of geometrically increasing intervals
   $\Omega := \left\{\ldots, \left[2^{-2/4}, 2^{-1/4}\right), \left[2^{-1/4}, 1\right), \left[1, 2^{1/4}\right), \left[2^{1/4}, 2^{2/4}\right), \ldots\right\} \cup \{[0, 0]\}$
5 Run $(\varepsilon_0, \delta_0)$-DP histogram learner of Lemma B.1 on $\{\psi_j\}_{j=1}^k$ over $\Omega$
6 **if** all the bins are empty **then** Return $\perp$
7 Let $[\ell, r]$ be a non-empty bin that contains the maximum number of points in the DP histogram
8 Return $\ell$

---

**Lemma F.1.** *Algorithm 3 is $(\varepsilon_0, \delta_0)$-DP. If $\{x_i\}_{i=1}^n$ are i.i.d. samples from $(K, a)$-sub-Weibull distributions with zero mean and covariance $\Sigma$ and*

$$n = \tilde{O}\left(\frac{\log^{2a}(1/(\delta_0 \zeta))}{\varepsilon_0}\right) \,,$$

*with a large enough constant then Algorithm 3 returns $\Gamma$ such that, with probability $1 - \zeta$,*

$$\frac{1}{\sqrt{2}} \operatorname{Tr}(\Sigma) \leq \Gamma \leq \sqrt{2} \operatorname{Tr}(\Sigma) \,.$$

We provide a proof in App. F.1.

### F.1 PROOF OF LEMMA F.1 ON THE PRIVATE NORM ESTIMATION

By Hanson-Wright inequality in Lemma J.1 and union bound, there exists constant $c > 0$ such that with probability $1 - \zeta$,

$$|\frac{1}{b} \sum_{i=1}^{b} \|x_i\|^2 - \text{Tr}(\Sigma)| \leq cK^2 \text{Tr}(\Sigma) \left( \sqrt{\frac{\log(1/\zeta)}{b}} + \frac{\log^{2a}(1/\zeta)}{b} \right) , \tag{18}$$

This means there exists a constant $c' > 0$ such that if $b \geq c'K^2 \log^{2a}(k/\zeta)$, then for all $j \in [k]$.

$$|\psi_j - \text{Tr}(\Sigma)| \leq 2^{1/8} \text{Tr}(\Sigma) \tag{19}$$

With probability $1 - \zeta$, $\{\psi_j\}_{j=1}^{k}$ lie in interval of size $2^{1/4} \text{Tr}(\Sigma)$. Thus, at most two consecutive bins are filled with $\{\psi_j\}_{j=1}^{k}$. Denote them as $I = I_1 \cup I_2$. Our analysis indicates that $\mathbb{P}(\psi_i \in I) \geq 0.99$. By private histogram in Lemma B.1, if $k \geq \log(1/(\delta\zeta))/\varepsilon$, $|\hat{p}_I - \tilde{p}_I| \leq 0.01$ where $\hat{p}_I$ is the empirical count on $I$ and $\tilde{p}_I$ is the noisy count on $I$. Under this condition, one of these two intervals are released. This results in multiplicative error of $\sqrt{2}$.

## G PROOF OF THE RESILIENCE IN LEMMA J.7

We apply following resilience property for general distribution characterized by Orlicz function from Zhu et al. (2019).

**Lemma G.1** ((Zhu et al., 2019, Theorem 3.4)). *Dataset $S = \{x_i \in \mathbb{R}^d\}_{i=1}^{n}$ consists i.i.d. samples from a distribution $\mathcal{D}$. Suppose $\mathcal{D}$ is zero mean and satisfies $\mathbb{E}_{x \sim \mathcal{D}} \left[ \psi \left( \frac{(v^\top x)^2}{\kappa^2 \mathbb{E}_{x \sim \mathcal{D}}[(v^\top x)^2]} \right) \right] \leq 1$ for all $v \in \mathbb{R}^d$, where $\psi(\cdot)$ is Orlicz function. Let $\Sigma = \mathbb{E}_{x \sim \mathcal{D}}[xx^\top]$. Suppose $\alpha \leq \bar{\alpha}$, where $\bar{\alpha}$ satisfies $(1 + \bar{\alpha}/2) \cdot 2\kappa^2 \bar{\alpha} \psi^{-1}(2/\bar{\alpha}) < 1/3$, $\bar{\alpha} \leq 1/4$. Then there exists constant $c_1, C_2$ such that if $n \geq c_1((d + \log(1/\zeta))/(\alpha^2))$, with probability $1 - \zeta$, for any $T \subset S$ of size $|T| \geq (1 - \alpha)n$, the following holds:*

$$\left\| \Sigma^{-1/2} \left( \frac{1}{|T|} \sum_{i \in T} x_i \right) \right\| \leq C_2 \kappa \alpha \sqrt{\psi^{-1}(1/\alpha)} \tag{20}$$

*and*

$$\left\| \mathbf{I}_d - \Sigma^{-1/2} \left( \frac{1}{|T|} \sum_{i \in T} x_i x_i^\top \right) \Sigma^{-1/2} \right\|_2 \leq C_2 \kappa^2 \alpha \psi^{-1}(1/\alpha) . \tag{21}$$

Let $\psi(t) = e^{t^{1/(2a)}}$. It is easy to see that $\psi(t)$ is a valid Orlicz function. Then if $x_i$ is $(K, a)$-sub-Weibull, then we know

$$\left\| \Sigma^{-1/2} \left( \frac{1}{|T|} \sum_{i \in T} x_i \right) \right\| \leq C_2 K \alpha \sqrt{\log^{2a}(1/\alpha)} , \tag{22}$$

and

$$\left\| \mathbf{I}_d - \Sigma^{-1/2} \left( \frac{1}{|T|} \sum_{i \in T} x_i x_i^\top \right) \Sigma^{-1/2} \right\|_2 \leq C_2 K^2 \alpha \log^{2a}(1/\alpha) . \tag{23}$$

This implies

$$(1 - C_2 K^2 \alpha \log^{2a}(1/\alpha))\mathbf{I}_d \preceq \Sigma^{-1/2} \left( \frac{1}{|T|} \sum_{i \in T} x_i x_i^\top \right) \Sigma^{-1/2} \preceq (1 + C_2 K^2 \alpha \log^{2a}(1/\alpha))\mathbf{I}_d . \tag{24}$$

Using the fact that $C^\top A C \preceq C^\top B C$ if $A \preceq B$, we know

$$(1 - C_2 K^2 \alpha \log^{2a}(1/\alpha))\Sigma \preceq \frac{1}{|T|} \sum_{i \in T} x_i x_i^\top \preceq (1 + C_2 K^2 \alpha \log^{2a}(1/\alpha))\Sigma \,. \tag{25}$$

This implies resilience properties of $x_i$ and $z_i$ in Eq. (8) and Eq. (9) in Definition C.1 respectively. Next, we show the resilience property of $x_i z_i$.

By $ab \le \frac{a^2}{2} + \frac{b^2}{2}$, for any fixed $v \in \mathbb{R}^d$,

$$\mathbb{E}[\exp\left(\left(\frac{|\langle x_i z_i, v\rangle|^2}{K^4 \sigma^2 v^\top \Sigma v}\right)^{1/(4a)}\right)] \le \mathbb{E}\left[\exp\left(\left(\frac{|\langle x_i, v\rangle|^2}{K^2 v^\top \Sigma v}\right)^{1/(2a)}/2\right)\exp\left(\left(\frac{z_i^2}{K^2 \sigma^2}\right)^{1/(2a)}/2\right)\right] \tag{26}$$

$$\le \frac{1}{2}\left(\mathbb{E}\left[\exp\left(\left(\frac{|\langle x_i, v\rangle|^2}{K^2 v^\top \Sigma v}\right)^{1/(2a)}\right)\right] + \mathbb{E}\left[\exp\left(\left(\frac{z_i^2}{K^2 \sigma^2}\right)^{1/(2a)}\right)\right]\right) \tag{27}$$

$$\le 1 \,. \tag{28}$$

Since $\mathbb{E}[x_i z_i] = 0$, (Zhu et al., 2019, Lemma E.3) implies that there exists constant $c_1, C_2 > 0$ such that if $n \ge c_1(d + \log(1/\zeta))/(\alpha^2)$, with probability $1 - \zeta$, for any $T \subset S_{\text{good}}$ of size $|T| \ge (1 - \alpha)n$,

$$\left\| \Sigma^{-1}\left(\frac{1}{|T|}\sum_{i \in T} x_i z_i\right)\right\| \le C_2 K^2 \sigma \alpha \log^{2a}(1/\alpha) \,. \tag{29}$$

# H   PROOF OF THEOREM 3 ON THE ANALYSIS OF ALGORITHM 1

The main theorem builds upon the following lemma that analyzes a (stochastic) gradient descent method, where the randomness is from the DP noise we add and the analysis only relies on certain deterministic conditions on the dataset including resilienece and concentration. Theorem 3 follows in a straightforward manner by collecting Theorem 4, Lemma F.1, Lemma 4.2, and Lemma H.1.

**Lemma H.1.** *Algorithm 1 is $(\varepsilon, \delta)$-DP. Under Assumptions 1 and 2 for any $\zeta \in (0, 1)$ and $\alpha \ge \alpha_{\text{corrupt}}$ satisfying $K^2 \alpha \log^{2a}(1/\alpha) \log(\kappa) \le c$ for some universal constant $c > 0$, if distance threshold is small enough such that*

$$\theta_t \le 3C_2^{1/2} K \log^a(1/(2\alpha)) \cdot (\|w^* - w_t\|_\Sigma + \sigma) \,, \tag{30}$$

*and large enough such that the number of clipped clean data points is no larger than $\alpha n$, at every round, the norm threshold is large enough such that*

$$\Theta \ge K\sqrt{\text{Tr}(\Sigma)}\log^a(n/\zeta) \,, \tag{31}$$

*and sample size is large enough such that*

$$n = O\left(K^2 d \log(d/\zeta)\log^{2a}(n/\zeta) + \frac{d + \log(1/\zeta)}{\alpha^2} + \frac{K^2 T^{1/2} d \log(T/\delta)\log^a(n/(\alpha\zeta))}{\varepsilon\alpha}\right) \,, \tag{32}$$

*with a large enough constant, then the choices of a step size, $\eta = 1/(C\lambda_{\max}(\Sigma))$ for some $C \ge 1.1$, and the number of iterations, $T = \tilde{\Theta}(\kappa \log(\|w^*\|))$, ensures that Algorithm 1 outputs $w_T$ satisfying the following with probability $1 - \zeta$:*

$$\mathbb{E}_{\nu_1, \cdots, \nu_t \sim \mathcal{N}(0, \mathbf{I}_d)}[\|w_T - w^*\|_\Sigma^2] \lesssim K^4 \sigma^2 \log^2(\kappa)\alpha^2 \log^{4a}(1/\alpha) \,, \tag{33}$$

*where the expectation is taken over the noise added for DP and $\tilde{\Theta}(\cdot)$ hides logarithmic terms in $K, \sigma, d, n, 1/\varepsilon, \log(1/\delta), 1/\alpha$.*

*Proof of Lemma H.1.* We first prove a set of deterministic conditions on the clean dataset, which is sufficient for the analysis of the gradient descent.

**Step 1: Sufficient deterministic conditions on the clean dataset.** Let $S_{\text{good}}$ be the uncorrupted dataset for $S_3$ and $S_{\text{bad}}$ be the corrupted datapoints in $S_3$. Let $G := S_{\text{good}} \cap S_3 = S_3 \setminus S_{\text{bad}}$ denote the clean data that remains in the input dataset. Let $\lambda_{\max} = \|\Sigma\|_2$. Define $\hat{\Sigma} := (1/n) \sum_{i \in G} x_i x_i^\top$, $\hat{B} := \mathbf{I}_d - \eta \hat{\Sigma}$. Lemma J.4 implies that if $n = O(K^2 d \log(d/\zeta) \log^{2a}(n/\zeta))$, then

$$0.9\Sigma \preceq \hat{\Sigma} \preceq 1.1\Sigma . \tag{34}$$

We pick step size $\eta$ such that $\eta \le 1/(1.1\lambda_{\max})$ to ensure that $\eta \le 1/\|\hat{\Sigma}\|_2$. Since the covariates $\{x_i\}_{i \in S}$ are not corrupted, from Lemma J.3, we know with probability $1 - \zeta$, for all $i \in S_3$,

$$\|x_i\|^2 \le K^2 \operatorname{Tr}(\Sigma) \log^{2a}(n/\zeta) . \tag{35}$$

Lemma J.7 implies that if $n = O((d + \log(1/\zeta))/(\alpha^2))$, then there exists a universal constant $C_2$ such that $S_3$ is, following Definition J.6, with respect to $(w^*, \Sigma, \sigma)$, $(\alpha_{\text{corrupt}}, \alpha, C_2 K^2 \alpha \log^{2a}(1/\alpha), C_2 K^2 \alpha \log^{2a}(1/\alpha), C_2 K^2 \alpha \log^{2a}(1/\alpha), C_2 K \alpha \log^a(1/\alpha))$-corrupt good. Such corrupt good sets have a sufficiently large, $1 - \alpha_{\text{corrupt}}$, fraction of points that satisfy a good property that we need: resilience. The rest of the proof is under Eq. (34), Eq. (35), and that $S_{\text{good}}$ is resilient.

**Step 2: Upper bounding the deterministic noise in the gradient.** In this step, we bound the deviation of the gradient from its mean. There are several sources of deviation: $(i)$ clipping, $(ii)$ adversarial corruptions, and $(iii)$ randomness of the data noise and privacy noise. We will show that deviations from all these sources can be controlled deterministically under the corrupt-goodness (i.e., resilience).

Let $\phi_t = (\sqrt{2\log(1.25/\delta_0)}\Theta\theta_t)/(\varepsilon_0 n)$, which ensures that we add enough noise to guarantee $(\varepsilon_0, \delta_0)$-DP for each step of gradient descent. This follows from the standard Gaussian mechanism in Lemma 2.2 and the fact that each gradient is clipped to the norm of $\Theta\theta_t$, resulting in a DP sensitivity of $\Theta\theta_t/n$. The fact that this sensitivity scales as $1/n$ is one of the main reasons for the performance gain we get over Varshney et al. (2022) that uses a minimatch of size $n/\kappa$ with sensitivity scaling as $\kappa/n$. Define $g_i^{(t)} := x_i(x_i^\top w_t - y_i)$. For $i \in S_{\text{good}}$, we know $y_i = x_i^\top w^* + z_i$. Let $\tilde{g}_i = \operatorname{clip}_\Theta(x_i)\operatorname{clip}_{\theta_t}(x_i^\top w_t - y_i)$. Note that under Eq. (35), $\operatorname{clip}_\Theta(x_i) = x_i$ for all $i \in S_3$.

From Algorithm 1, we can write one-step update rule as follows:

$$
\begin{aligned}
& w_{t+1} - w^* \\
={}& w_t - \eta\left(\frac{1}{n}\sum_{i \in S}\tilde{g}_i^{(t)} + \phi_t\nu_t\right) - w^* \\
={}& \left(\mathbf{I} - \frac{\eta}{n}\sum_{i \in G}x_i x_i^\top\right)(w_t - w^*) + \frac{\eta}{n}\sum_{i \in G}x_i z_i + \frac{\eta}{n}\sum_{i \in G}(g_i^{(t)} - \tilde{g}_i^{(t)}) - \eta\phi_t\nu_t - \frac{\eta}{n}\sum_{i \in S_{\text{bad}}}\tilde{g}_i^{(t)}
\end{aligned}
\tag{36}
$$

Let $E_t := \{i \in G : \theta_t \le |x_i^\top w_t - y_i|\}$ be the set of clipped clean data points such that $\sum_{i \in G}(g_i^{(t)} - \tilde{g}_i^{(t)}) = \sum_{i \in E_t}(g_i^{(t)} - \tilde{g}_i^{(t)})$. We define $\hat{v} := (1/n)\sum_{i \in G}x_i z_i$, $u_t^{(1)} := (1/n)\sum_{i \in E_t}x_i x_i^\top(w_t - w^*)$, $u_t^{(2)} := (1/n)\sum_{i \in E_t}-x_i z_i$, and $u_t^{(3)} := (1/n)\sum_{i \in S_{\text{bad}} \cup E_t}\tilde{g}_i^{(t)}$.

We can further write the update rule as:

$$w_{t+1} - w^* = \hat{B}(w_t - w^*) + \eta\hat{v} + \eta u_{t-1}^{(1)} + \eta u_{t-1}^{(2)} - \eta\phi_t\nu_t - \eta u_{t-1}^{(3)} . \tag{37}$$

We bound each term one-by-one. Since $G \subset S_{\text{good}}$ and $|G| = (1 - \alpha_{\text{corrupt}})n$, using the resilience property in Eq. (7), we know

$$
\begin{aligned}
\|\Sigma^{-1/2}\hat{v}\| &= (1 - \alpha_{\text{corrupt}})\max_{\|v\|=1}\Sigma^{-1/2}\left\langle v, \frac{1}{(1 - \alpha_{\text{corrupt}})n}\sum_{i \in G}x_i z_i\right\rangle \\
&\le (1 - \alpha_{\text{corrupt}})C_2 K^2\alpha\log^{2a}(1/\alpha)\sigma \tag{38} \\
&\le C_2 K^2\alpha\log^{2a}(1/\alpha)\sigma . \tag{39}
\end{aligned}
$$

Let $\tilde{\alpha} = |E_t|/n$. By assumption, we know $\tilde{\alpha} \leq \alpha$ (which holds for the given dataset due to Lemma 4.2), and

$$\|\Sigma^{-1/2} u_t^{(1)}\| = \|\Sigma^{-1/2} \frac{1}{n} \sum_{i \in E_t} x_i x_i^\top (w_t - w^*)\| .$$

From Corollary J.8, we know

$$\left| \|\Sigma^{-1/2} \frac{1}{|E_t|} \sum_{i \in E_t} x_i x_i^\top (w_t - w^*)\| - \|w_t - w^*\|_\Sigma \right|$$

$$= \left| \max_{u:\|u\|=1} \frac{1}{|E_t|} \sum_{i \in E_t} u^\top \Sigma^{-1/2} x_i x_i^\top (w_t - w^*)\| - \max_{v:\|v\|=1} v^\top \Sigma^{1/2}(w_t - w^*) \right|$$

$$\leq \max_{u:\|u\|=1} \left| \frac{1}{|E_t|} \sum_{i \in E_t} u^\top \Sigma^{-1/2} x_i x_i^\top \Sigma^{-1/2} \Sigma^{1/2}(w_t - w^*)\| - u^\top \Sigma^{1/2}(w_t - w^*) \right|$$

$$\leq \max_{u:\|u\|=1} \left| \frac{1}{|E_t|} \sum_{i \in E_t} u^\top \left( \Sigma^{-1/2} x_i x_i^\top \Sigma^{-1/2} - \mathbf{I}_d \right) \Sigma^{1/2}(w_t - w^*)\| \right|$$

$$= \left\| \frac{1}{|E_t|} \sum_{i \in E_t} \left( \Sigma^{-1/2} x_i x_i^\top \Sigma^{-1/2} - \mathbf{I}_d \right) \Sigma^{1/2}(w_t - w^*) \right\|$$

$$\leq \left\| \frac{1}{|E_t|} \sum_{i \in E_t} \left( \Sigma^{-1/2} x_i x_i^\top \Sigma^{-1/2} - \mathbf{I}_d \right) \right\| \cdot \left\| \Sigma^{1/2}(w_t - w^*) \right\|$$

$$\leq \frac{2 - \tilde{\alpha}}{\tilde{\alpha}} C_2 K^2 \alpha \log^{2a}(1/\alpha) \|w_t - w^*\|_\Sigma .$$

This implies that

$$\|\Sigma^{-1/2} u_t^{(1)}\| \leq \|\Sigma^{-1/2} \frac{1}{n} \sum_{i \in E} x_i x_i^\top (w_t - w^*)\|$$

$$\leq \left( \tilde{\alpha} + 2C_2 K^2 \alpha \log^{2a}(1/\alpha) \right) \|w_t - w^*\|_\Sigma$$

$$\leq 3C_2 K^2 \alpha \log^{2a}(1/\alpha) \|w_t - w^*\|_\Sigma , \tag{40}$$

where the last inequality follows from the fact that $\tilde{\alpha} \leq \alpha$ and our assumption that $C_2 K^2 \log^{2a}(1/\bar{\alpha}) \geq 1$ from Assumption 2. Similarly, we use resilience property in Eq. (7) instead of Eq. (8), we can show that

$$\|\Sigma^{-1/2} u_t^{(2)}\| \leq 3C_2 K^2 \alpha \log^{2a}(1/\alpha)\sigma . \tag{41}$$

Next, we consider $u_t^{(3)}$. Since $|S_{\text{bad}}| \leq \alpha_{\text{corrupt}} n$ and $|E_t| \leq \alpha n$, using Eq. (10) and Corollary J.8, we have

$$\|\Sigma^{-1/2} u_t^{(3)}\| = \max_{v:\|v\|=1} \frac{1}{n} \sum_{i \in S_{\text{bad}} \cup E_t} v^\top \Sigma^{-1/2} x_i \text{clip}_{\theta_t}(x_i^\top w_t - y_i)$$

$$\leq 2C_2 K \alpha \log^a(1/\alpha)\theta_t$$

$$\leq 6C_2^{1.5} K^2 \alpha \log^{2a}(1/\alpha)(\|w_t - w^*\|_\Sigma + \sigma) . \tag{42}$$

Now we use Eq. (39), Eq. (40), Eq. (41) and Eq. (42) to bound the final error from update rule in Eq. (37).

**Step 3: Analysis of the $t$-steps recurrence relation.** We have controlled the deterministic noise in the last step. In this step, we will upper bound the noise introduced by the Gaussian noise for the purpose of privacy, and show the expected distance to optimum decrease every step.

Define $u_t = (\hat{v} + u_t^{(1)} + u_t^{(2)} - u_t^{(3)})$. We can rewrite Eq. (37) as

$$w_{t+1} - w^* = \hat{B}(w_t - w^*) + \eta u_t - \eta \phi_t \nu_t \tag{43}$$

$$= \hat{B}^{t+1}(w_0 - w^*) + \eta \sum_{i=0}^{t} \hat{B}^i u_{t-i} - \eta \sum_{i=0}^{t} \phi_{t-i} \hat{B}^i \nu_{t-i} . \tag{44}$$

Taking expectations of $\hat{\Sigma}$-norm square with respect to $\nu_1, \cdots, \nu_t$, we have

$$\mathbb{E}_{\nu_1,\ldots,\nu_t \sim \mathcal{N}(0, \mathbf{I}_d)} \|w_{t+1} - w^*\|_{\hat{\Sigma}}^2 \tag{45}$$

$$\leq 2\|\hat{B}^{t+1}(w_0 - w^*)\|_{\hat{\Sigma}}^2 + 2\mathbb{E}[\|\eta \sum_{i=0}^{t} \hat{B}^i u_{t-i}\|_{\hat{\Sigma}}^2] + \eta^2 \sum_{i=0}^{t} \mathrm{Tr}(\hat{B}^{2i}\hat{\Sigma})\mathbb{E}[\phi_{t-i}^2] \tag{46}$$

$$\leq 2\|\hat{B}^{t+1}(w_0 - w^*)\|_{\hat{\Sigma}}^2 + 2\eta^2 \mathbb{E}[\sum_{i=0}^{t}\sum_{j=0}^{t} \|\hat{B}^i u_{t-i}\|_{\hat{\Sigma}}\|\hat{B}^j u_{t-j}\|_{\hat{\Sigma}}] \tag{47}$$

$$+ \eta^2 \sum_{i=0}^{t} \mathrm{Tr}(\hat{B}^{2i}\hat{\Sigma})\mathbb{E}[\phi_{t-i}^2] , \tag{48}$$

where at the second step we used the fact that $\nu_1, \nu_2, \cdots, \nu_t$ are independent isotropic Gaussian.

Note that

$$\eta\|\hat{B}^i u_{t-i}\|_{\hat{\Sigma}} = \eta\|\hat{\Sigma}^{1/2}\hat{B}^i\hat{\Sigma}^{1/2}\hat{\Sigma}^{-1/2}u_{t-i}\|$$
$$\leq \eta\|\hat{\Sigma}^{1/2}\hat{B}^i\hat{\Sigma}^{1/2}\|_2 \cdot \|\hat{\Sigma}^{-1/2}u_{t-i}\|$$
$$\leq \eta\|\hat{\Sigma}^{1/2}\hat{B}^i\hat{\Sigma}^{1/2}\|_2 \hat{\rho}(\alpha)\left(\|w_{t-i} - w^*\|_{\hat{\Sigma}} + \sigma\right)$$
$$\leq \frac{1}{i+1}\hat{\rho}(\alpha)\left(\|w_{t-i} - w^*\|_{\hat{\Sigma}} + \sigma\right) ,$$

where $\hat{\rho}(\alpha) = 1.1(6C_2 + 6C_2^{1.5})K^2\alpha \log^{2a}(1/\alpha)$, and the second inequality follows from Eq. (40), Eq. (41), Eq. (42) and the deterministic condition in Eq. (34). Note that the last inequality is true because $\eta \leq 1/(1.1\lambda_{\max})$ and $\|\hat{\Sigma}^{1/2}\hat{B}^i\hat{\Sigma}^{1/2}\|_2 \leq \|\mathbf{I}_d - \eta\hat{\Sigma}\|_2^i\|\hat{\Sigma}\|_2 \leq \lambda_{\max}/(i+1) .$

This implies

$$\mathbb{E}[\eta^2 \sum_{i=0}^{t}\sum_{j=0}^{t} \|\hat{B}^i u_{t-i}\|_{\hat{\Sigma}}\|\hat{B}^j u_{t-j}\|_{\hat{\Sigma}}] \tag{49}$$

$$\leq 4\,\mathbb{E}[\sum_{i=0}^{t}\sum_{j=0}^{t} \frac{\hat{\rho}(\alpha)^2}{(i+1)(j+1)}(\mathbb{E}[\|w_{t-i} - w^*\|_{\hat{\Sigma}}^2] + \mathbb{E}[\|w_{t-j} - w^*\|_{\hat{\Sigma}}^2] + \sigma^2) \tag{50}$$

$$\leq 8(\sum_{i=0}^{t} \frac{1}{i+1})^2\hat{\rho}(\alpha)^2(\max_i \mathbb{E}[\|w_{t-i} - w^*\|_{\hat{\Sigma}}^2] + \sigma^2) \tag{51}$$

$$\leq 8(\log t)^2\hat{\rho}(\alpha)^2(\max_i \mathbb{E}[\|w_{t-i} - w^*\|_{\hat{\Sigma}}^2] + \sigma^2) , \tag{52}$$

Then,

$$\|\hat{B}^{t+1}(w_0 - w^*)\|_{\hat{\Sigma}}^2 = \|\hat{\Sigma}^{1/2}\hat{B}^{t+1}\hat{\Sigma}^{-1/2}\hat{\Sigma}^{1/2}(w_0 - w^*)\|^2$$
$$\leq (1 - \frac{1}{\kappa})^{2(t+1)}\|w_0 - w^*\|_{\hat{\Sigma}}^2 \leq e^{-2(t+1)/\kappa}\|w_0 - w^*\|_{\hat{\Sigma}}^2 ,$$

and for $n \gtrsim (1/\varepsilon)\sqrt{\kappa d \log(1/\delta)/\alpha}$,

$$\eta^2 \sum_{i=0}^{t} \text{Tr}(\hat{B}^{2i}\hat{\Sigma})\mathbb{E}[\phi_{t-i}^2] \tag{53}$$

$$\leq \eta^2 \sum_{i=0}^{t} \|\mathbf{I}_d - \eta\hat{\Sigma}\|_2^{2i}\|\hat{\Sigma}\|_2 \cdot \frac{2\log(1.25/\delta_0)K^2 \text{Tr}(\Sigma)\log^{2a}(n/\zeta_0)C_2 K^2 \log^{2a}(1/(2\alpha))(\mathbb{E}[\|w_{t-i}-w^*\|_{\hat{\Sigma}}^2] + \sigma^2)}{\varepsilon_0^2 n^2} \tag{54}$$

$$\leq 4 \sum_{i=0}^{t} (\frac{1}{i+1})^2 \hat{\rho}(\alpha)^2 (\mathbb{E}[\|w_{t-i}-w^*\|_{\hat{\Sigma}}^2] + \sigma^2) . \tag{55}$$

We have

$$\mathbb{E}_{\nu_1,\ldots,\nu_t \sim \mathcal{N}(0,\mathbf{I}_d)}[\|w_{t+1}-w^*\|_{\hat{\Sigma}}^2] \leq 2e^{-2(t+1)/\kappa}\|w_0-w^*\|_{\hat{\Sigma}}^2 + 20(\log t)^2\hat{\rho}(\alpha)^2(\max_{i\in[t]}\mathbb{E}[\|w_{t-i}-w^*\|_{\hat{\Sigma}}^2] + \sigma^2) .$$

Note that this also implies that

$$\mathbb{E}[\|(w_{t'+t}-w^*)\|_{\hat{\Sigma}}^2|w_{t'}] \leq 2e^{-2t/\kappa}\|w_{t'}-w^*\|_{\hat{\Sigma}}^2 + 20\hat{\rho}(\alpha)^2 \sum_{i=0}^{t-1}(\frac{1}{i+1})^2(\mathbb{E}[\|w_{t'+t-i}-w^*\|_{\hat{\Sigma}}^2|w_{t'}] + \sigma^2) , \tag{56}$$

which implies

$$\mathbb{E}[\|(w_{t'+t}-w^*)\|_{\hat{\Sigma}}^2] \leq 2e^{-2t/\kappa}\mathbb{E}[\|w_{t'}-w^*\|_{\hat{\Sigma}}^2] + 20\hat{\rho}(\alpha)^2 \sum_{i=0}^{t-1}(\frac{1}{i+1})^2(\mathbb{E}[\|w_{t'+t-i}-w^*\|_{\hat{\Sigma}}^2] + \sigma^2) \tag{57}$$

$$\leq 2e^{-2t/\kappa}\mathbb{E}[\|w_{t'}-w^*\|_{\hat{\Sigma}}^2] + 20(\log t)^2\hat{\rho}(\alpha)^2(\max_{i\in[t]}\mathbb{E}[\|w_{t'+t-i}-w^*\|_{\hat{\Sigma}}^2] + \sigma^2) \tag{58}$$

**Step 4: End-to-end analysis of the convergence.** In the last step, we shown that the amount of estimation error decrease depends on the estimation error of the previous $t$ steps. In order for the estimation error to decrease by a constant factor, we will take $t = \kappa$. Roughly speaking, we will prove that for every $\kappa$ steps, the estimation error will decrease by a constant factor, if it is much larger than $O((\log \kappa)^2\hat{\rho}(\alpha)^2\sigma^2)$. This implies we will reach $O((\log \kappa)^2\hat{\rho}(\alpha)^2\sigma^2)$ error with in $\tilde{O}(\kappa)$ steps.

For any integer $s \geq 0$, as long as $\max_{i\in[(s-1)\kappa+1,s\kappa]}\mathbb{E}[\|w_i-w^*\|_{\hat{\Sigma}}^2] \geq 2(\log \kappa)^2\hat{\rho}(\alpha)^2\sigma^2$,

$$\max_{i\in[s\kappa+1,(s+1)\kappa]}\mathbb{E}[\|w_i-w^*\|_{\hat{\Sigma}}^2] \leq (\frac{1}{e^2} + (\log \kappa)^2\hat{\rho}(\alpha)^2)\max_{i\in[(s-1)\kappa+1,s\kappa]}\mathbb{E}[\|w_i-w^*\|_{\hat{\Sigma}}^2] + (\log 2\kappa)^2\hat{\rho}(\alpha)^2\sigma^2 . \tag{59}$$

Assuming $\hat{\rho}(\alpha)^2(\log \kappa)^2 \leq 1/2 - 1/e^2$, the maximum expected error in a length $\kappa$ sequence decrease by a factor of $1/2$ every time.

Now we bound the maximum expected error in the first length $\kappa$ sequence: $\max_{i\in[0,\kappa-1]}\mathbb{E}[\|w_i-w^*\|_{\hat{\Sigma}}^2]$. Since

$$\mathbb{E}[\|w_i-w^*\|_{\hat{\Sigma}}^2] \leq e^{-2i/\kappa}\|w_0-w^*\|_{\hat{\Sigma}}^2 + (\log i)^2\hat{\rho}(\alpha)^2\max_{j\in[0,i-1]}\mathbb{E}[\|w_j-w^*\|_{\hat{\Sigma}}^2] + (\log i)^2\hat{\rho}(\alpha)^2\sigma^2 .$$

As a function of $i$, $\max_{j\in[0,i-1]}\mathbb{E}[\|w_j-w^*\|_{\hat{\Sigma}}^2]$ only increase when it is smaller than

$$\frac{1}{1-(\log i)^2\hat{\rho}(\alpha)^2}(\|w_0-w^*\|_{\hat{\Sigma}}^2 + (\log i)^2\hat{\rho}(\alpha)^2\sigma^2) .$$

Thus we conclude

$$\max_{i\in[0,\kappa-1]}\mathbb{E}[\|w_i-w^*\|_{\hat{\Sigma}}^2] \leq \frac{1}{1-(\log \kappa)^2\hat{\rho}(\alpha^2)}(\|w_0-w^*\|_{\hat{\Sigma}}^2 + (\log \kappa)^2\hat{\rho}(\alpha^2)\sigma^2)$$

$s = \log(\|w^*\|/(\hat{\rho}(\alpha)\sigma))$ will give us

$$\mathbb{E}[\|w_{s\kappa+1}-w^*\|_{\hat{\Sigma}}^2] \leq (\log \kappa)^2\hat{\rho}(\alpha)^2\sigma^2 .$$

$\square$

# I  LOWER BOUNDS

## I.1  PROOF OF PROPOSITION 3.1 FOR LABEL CORRUPTION LOWER BOUNDS

We first prove the following lemma.

**Lemma I.1.** *Consider an $\alpha$ label-corrupted dataset $S = \{(x_i, y_i)\}_{i=1}^n$ with $\alpha < 1/2$, that is generated from either $x_i \sim \mathcal{N}(0, 1), y_i \sim \mathcal{N}(0, 1)$ or $x_i \sim \mathcal{N}(0, 1), z_i \sim \mathcal{N}(0, 1 - \alpha^2), y_i = \alpha x_i + z_i$. It is impossible to distinguish the two hypotheses with probability larger than $1/2$.*

In the first case,

$$(x_i, y_i) \sim \mathcal{P}_1 = \mathcal{N}(0, \begin{bmatrix} 1 & 0 \\ 0 & 1 \end{bmatrix}).$$

In the second case,

$$(x_i, y_i) \sim \mathcal{P}_2 = \mathcal{N}(0, \begin{bmatrix} 1 & \alpha \\ \alpha & 1 \end{bmatrix}).$$

By simple calculation, it holds that $D_{KL}(\mathcal{P}_1 || \mathcal{P}_2) = -\frac{1}{2} \log(1 - \alpha^2) \leq \alpha^2/2$ for all $\alpha < 1/2$. Then, Pinsker's inequality implies that $D_{TV}(\mathcal{P}_1 || \mathcal{P}_2) \leq \alpha/2$. Since the covariate $x_i$ follows from the same distribution in the two cases, and the total variation distance between the two cases is less than $\alpha/2$. This means there is an label corruption adversary that change $\alpha/2$ fraction of $y_i$'s in $\mathcal{P}_1$ to make it identical to $\mathcal{P}_2$. Therefore, no algorithm can distinguish the two cases with probability better than $1/2$ under $\alpha$ fraction of label corruption.

Since $\Sigma = 1$, $\sigma^2 \in [3/4, 1]$, the first case above has $w^* = 0$, and the second case has $w^* = \alpha$, this implies that no algorithm is able to achieve $\mathbb{E}[\|\hat{w} - w^*\|_\Sigma] < \sigma\alpha$ for all instances with $\|w^*\| \leq 1$ under $\alpha$ fraction of label corruption.

# J  TECHNICAL LEMMAS

**Lemma J.1** (Hanson-Wright inequality for subWeibull distributions Sambale (2020))**.** *Let $S = \{x_i \in \mathbb{R}^d\}_{i=1}^n$ be a dataset consist of i.i.d. samples from $(K, a)$-subWeibull distributions, then*

$$\mathbb{P}\left(\left|\frac{1}{n}\sum_{i=1}^n \|x_i\|^2 - \mathrm{Tr}(\Sigma)\right| \geq t\right) \leq 2\exp\left(-\min\left\{\frac{nt^2}{K^4(\mathrm{Tr}(\Sigma))^2}, \left(\frac{nt}{K^2\,\mathrm{Tr}(\Sigma)}\right)^{\frac{1}{2a}}\right\}\right). \quad (60)$$

**Lemma J.2.** *Let $Y \sim \mathrm{Lap}(b)$. Then for all $h > 0$, we have $\mathbb{P}(|Y| \geq hb) = e^{-h}$.*

**Lemma J.3.** *If $x \in \mathbb{R}^d$ is $(K, a)$-subWeibull for some $a \in [1/2, \infty)$. Then*

- *for any fixed $v \in \mathbb{R}^d$, with probability $1 - \zeta$,*

$$\langle x, v \rangle^2 \leq K^2 v^\top \Sigma v \log^{2a}(1/\zeta). \quad (61)$$

- *with probability $1 - \zeta$,*

$$\|x\|^2 \leq K^2 \mathrm{Tr}(\Sigma) \log^{2a}(1/\zeta). \quad (62)$$

We provide a proof in Appendix J.1.1.

**Lemma J.4.** *Dataset $S = \{x_i \in \mathbb{R}^d\}_{i=1}^n$ consists i.i.d. samples from a zero mean distribution $\mathcal{D}$. Suppose $\mathcal{D}$ is $(K, a)$-subWeibull. Define $\Sigma = \mathbb{E}_{x \sim \mathcal{D}}[xx^\top]$. Then there exists a constant $c_1 > 0$ such that with probability $1 - \zeta$,*

$$\left\|\frac{1}{n}\sum_{i=1}^n x_i x_i^\top - \Sigma\right\| \leq c_1 \left(\frac{K^2 d \log(d/\zeta) \log^{2a}(n/\zeta)}{n} + \sqrt{\frac{K^2 d \log(d/\delta) \log^{2a}(n/\zeta)}{n}}\right) \|\Sigma\|_2. \quad (63)$$

**Lemma J.5** (Lemma F.1 from Liu et al. (2022a))**.** *Let $x \in \mathbb{R}^d \sim \mathcal{N}(0, \Sigma)$. Then there exists universal constant $C_6$ such that with probability $1 - \zeta$,*

$$\|x\|^2 \leq C \mathrm{Tr}(\Sigma) \log(1/\zeta). \quad (64)$$

**Definition J.6** (Corrupt good set). *We say a dataset $S$ is $(\alpha_{\text{corrupt}}, \alpha, \rho_1, \rho_2, \rho_3, \rho_4)$-corrupt good with respect to $(w^*, \Sigma, \sigma)$ if it is $\alpha_{\text{corrupt}}$-corruption of an $(\alpha, \rho_1, \rho_2, \rho_3, \rho_4)$-resilient dataset $S_{\text{good}}$.*

**Lemma J.7.** *Under Assumptions 1 and 2, there exists positive constants $c_1$ and $C_2$ such that if $n \geq c_1((d + \log(1/\zeta))/\alpha^2$, then with probability $1 - \zeta$, $S_{\text{good}}$ is, with respect to $(w^*, \Sigma, \sigma)$, $(\alpha, C_2 K^2 \alpha \log^{2a}(1/\alpha), C_2 K^2 \alpha \log^{2a}(1/\alpha), C_2 K^2 \alpha \log^{2a}(1/\alpha), C_2 K \alpha \log^a(1/\alpha))$-resilient.*

We provide a proof in Appendix G.

**Corollary J.8** (Lemma 10 from Steinhardt et al. (2017) and Lemma 25 from Liu et al. (2022b)). *For a $(\alpha, \rho_1, \rho_2, \rho_3, \rho_4)$-resilient set $S$ with respect to $(w^*, \Sigma, \gamma)$ and any $0 \leq \tilde{\alpha} \leq \alpha$, the following holds for any subset $T \subset S$ of size at least $\tilde{\alpha}n$ and for any unit vector $v \in \mathbb{R}^{\bar{d}}$:*

$$\left| \frac{1}{|T|} \sum_{(x_i, y_i) \in T} \langle v, x_i \rangle (y_i - x_i^\top w^*) \right| \leq \frac{2 - \tilde{\alpha}}{\tilde{\alpha}} \rho_1 \sqrt{v^\top \Sigma v} \, \sigma \,, \tag{65}$$

$$\left| \frac{1}{|T|} \sum_{x_i \in T} \langle v, x_i \rangle^2 - v^\top \Sigma v \right| \leq \frac{2 - \tilde{\alpha}}{\tilde{\alpha}} \rho_2 v^\top \Sigma v \,, \tag{66}$$

$$\left| \frac{1}{|T|} \sum_{(x_i, y_i) \in T} (y_i - x_i^\top w^*)^2 - \sigma^2 \right| \leq \frac{2 - \tilde{\alpha}}{\tilde{\alpha}} \rho_3 \sigma^2 \,, \quad and \tag{67}$$

$$\left| \frac{1}{|T|} \sum_{x_i \in T} \langle v, x_i \rangle \right| \leq \frac{2 - \tilde{\alpha}}{\tilde{\alpha}} \rho_4 \sqrt{v^\top \Sigma v} \,. \tag{68}$$

### J.1 PROOF OF TECHNICAL LEMMAS

#### J.1.1 PROOF OF LEMMA J.3

Using Markov inequality,

$$\mathbb{P}\left( \langle v, x \rangle^2 \geq t^2 \right) = \mathbb{P}\left( e^{\langle v, x \rangle^{1/a}} \geq e^{t^{1/a}} \right) \tag{69}$$

$$\leq e^{-t^{1/a}} \mathbb{E}[e^{\langle v, x \rangle^{1/a}}] \tag{70}$$

$$\leq e^{-t^{1/a}} e^{K(\mathbb{E}[\langle v, x \rangle^2])^{1/(2a)}} \tag{71}$$

$$= \exp\left( -\left( \frac{t^2}{K^2 \mathbb{E}[\langle v, x \rangle^2]} \right)^{1/(2a)} \right) . \tag{72}$$

This implies for any fixed $v$, with probability $1 - \zeta$,

$$\langle x, v \rangle^2 \leq K^2 v^\top \mathbb{E}[xx^\top] v \log^{2a}(1/\zeta) \,. \tag{73}$$

For $j$-th coordinate, let $v = e_j$ where $j \in [d]$. Definition 2.1 implies

$$\mathbb{E}\left[ \exp\left( \left( \frac{x_j^2}{K^2 \operatorname{Tr}(\Sigma)} \right)^{1/(2a)} \right) \right] \leq \mathbb{E}\left[ \exp\left( \left( \frac{x_j^2}{K^2 \Sigma_{jj}} \right)^{1/(2a)} \right) \right] \leq 1 \,. \tag{74}$$

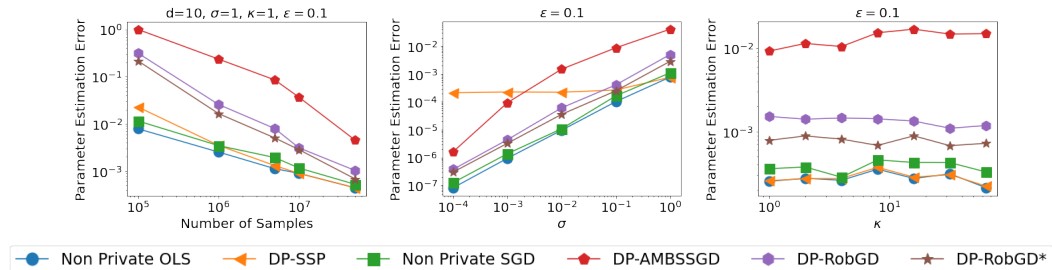

Figure 2: Performance of various techniques on DP linear regression. $d = 10$ in all the experiments. $n = 10^7, \kappa = 1$ in the $2^{nd}$ experiment. $n = 10^7, \sigma = 1$ in the $3^{rd}$ experiment.

Note that $f(x) = x^\alpha$ is concave function for $\alpha \leq 1$ and $x > 0$. Then $(a_1 + \cdots a_k)^\alpha \leq a_1^\alpha + \cdots a_k^\alpha$ holds for any positive numbers $a_1, \cdots, a_k > 0$. By our assumption that $1/(2a) \leq 1.$ , we have

$$\mathbb{E}[\exp\left(\left(\frac{\|x\|^2}{K^2 \operatorname{Tr}(\Sigma)}\right)^{1/(2a)}\right)] = \mathbb{E}[\exp\left(\left(\frac{x_1^2 + x_2^2 + \cdots + x_d^2}{K^2 \operatorname{Tr}(\Sigma)}\right)^{1/(2a)}\right)] \tag{75}$$

$$\leq \mathbb{E}[\prod_{j=1}^{d} \exp\left(\left(\frac{x_j^2}{K^2 \operatorname{Tr}(\Sigma)}\right)^{1/(2a)}\right)] \tag{76}$$

$$\leq \left(\frac{\sum_{j=1}^{d} \mathbb{E}[\exp\left(\left(\frac{x_j^2}{K^2 \operatorname{Tr}(\Sigma)}\right)^{1/(2a)}\right)]}{d}\right)^d \tag{77}$$

$$\leq 1 . \tag{78}$$

By Markov inequality,

$$\mathbb{P}\left(\|x\| \geq t\right) = \mathbb{P}\left(e^{\|x\|^{1/a}} \geq e^{t^{1/a}}\right) \tag{79}$$

$$\leq e^{-t^{1/a}} \mathbb{E}[e^{\|x\|^{1/a}}] \tag{80}$$

$$\leq \exp\left(-\left(\frac{t^2}{K^2 \operatorname{Tr}(\Sigma)}\right)^{1/(2a)}\right) . \tag{81}$$

This implies with probability $1 - \zeta$,

$$\|x\|^2 \leq K^2 \operatorname{Tr}(\Sigma) \log^{2a}(1/\zeta) . \tag{82}$$

## K    EXPERIMENTS

### K.1    DP LINEAR REGRESSION

Experimental results for $\epsilon = 0.1$ can be found in Figure 2. The observations are similar to the $\epsilon = 1$ case. In particular, DP-SSP has poor performance when $\sigma$ is small. In other settings, DP-SSP has better performance than DP-ROBGD.

### K.2    DP ROBUST LINEAR REGRESSION

In this section, we consider a stronger adversary for DP-ROBGD than the one considered in Section 5. Recall, for the adversary model considered in Section 5, DP-ROBGD was able to consistently estimate the parameter $w^*$ (*i.e.*, the parameter recovery error goes down to 0 as $n \to \infty$). This is because the algorithm was able to easily identify the corruptions and ignore the corresponding points while performing gradient descent. We now construct a different instance where the corruptions are hard to identify. Consequently, DP-ROBGD can no longer be consistent against the adversary.

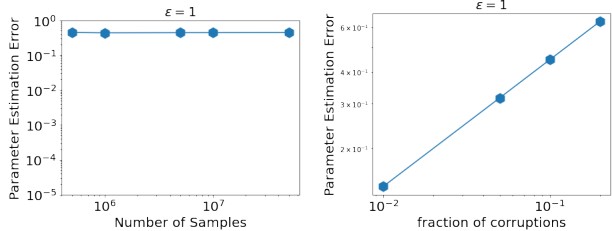

Figure 3: Performance against the stronger adversary

This hard instance is inspired by the lower bound in Bakshi & Prasad (2021) (see Theorem 6.1 of Bakshi & Prasad (2021)). This is a 2 dimensional problem where the first covariate is sampled uniformly from $[-1, 1]$. The second covariate, which is uncorrelated from the first, is sampled from a distribution with the following pdf

$$p(x^{(2)}) = \begin{cases} \frac{\alpha}{2} & \text{if } x^{(2)} \in \{-1, 1\} \\ \frac{1-\alpha}{2\alpha\sigma} & \text{if } x^{(2)} \in [-\sigma, \sigma] \\ 0 & \text{otherwise} \end{cases} .$$

We set $\sigma = 0.1$ in our experiments. The noise $z_i$ is sampled uniformly from $[-\sigma, \sigma]$. We consider two possible parameter vectors $w^* = (1, 1)$ and $w^* = (1, -1)$. It can be shown that the total variation (TV) distance between these problem instances (each parameter vector corresponds to one problem instance) is $\Theta(\alpha)$ (Bakshi & Prasad, 2021). What this implies is that, one can corrupt at most $\alpha$ fraction of the response variables and convert one problem instance into another. Since the distance (in $\Sigma$ norm) between the two parameter vectors is $\Omega(\alpha\sigma)$, any algorithm will suffer an error of $\Omega(\alpha\sigma)$.

We generate $10^7$ samples from this problem instance and add corruptions that convert one problem instance to the other. Figure 3 presents the results from this experiment. It can be seen that our algorithm works as expected. In particular, it is not consistent in this setting. Moreover, the parameter recovery error increases with the fraction of corruptions.

