# OpenReview forum: "Near Optimal Private and Robust Linear Regression"
_ICLR.cc/2023/Conference — Submitted to ICLR 2023_

### Official Review · Reviewer_p2zY · 2022-10-22

**Confidence:** 3
**Correctness:** 4
**Technical Novelty And Significance:** 3
**Empirical Novelty And Significance:** 3
**Recommendation:** 6

**Clarity, Quality, Novelty And Reproducibility:**

The result is clear and well-presented. There are various typos in the text which should be fixed.

**Strength And Weaknesses:**

At a technical level, the authors analyze DP-SGD for the linear regression problem (without an adversary) using two additional ideas. First, the work of Varshney et al. (2022) proposes a single pass approach while, in this work, a full-batch gradient descent algorithm is employed. This gives an improvement in the sample complexity. Second, adaptive clipping is used in order to ensure robustness against label-corruption.

In general, the paper is interesting and the result is nice. The  proposed private algorithm attains an improved sample complexity guarantee while being computationally efficient and robust to label noise. My main concern deals with the issue of technical novelty; the paper relies on standard techniques for achieving privacy (e.g., private histograms) and robustness (e.g., resilience). While the clipping idea is nice, it is essentially ad-hoc for label noise and cannot handle corruption in the covariates. Nevertheless, I believe that the contribution is sufficient for acceptance.

The paper is mostly well-written (there are various typos that can be fixed) and the results and previous work are clearly presented.


**Summary Of The Paper:**

This paper studies linear regression from i.i.d. examples under $(\epsilon, \delta)$-differential privacy in the settings (i) where there is no adversary and (ii) when a fraction of response variables are adversarially corrupted. While there exists a clear understanding about the optimal (inefficient) private and robust algorithm that solves this problem (namely the HPTR algorithm), it is not known whether this optimal sample complexity can be attained by a computationally efficient algorithm. Hence, this paper aims to improve on the number of samples required by the previous best-known algorithm by Varshney et al. (2022). This improvement is obtained by DP-SGD with some additional ideas.


**Summary Of The Review:**

The authors provide an efficient private and robust algorithm for linear regression, improving on the previous sample complexity bound. I believe that the contribution is beyond the acceptance threshold.

---

> ### Author Response · Authors · 2022-11-12
> **Thank you for your insightful feedback**
>
> We want to clarify two contributions of our paper that we think are important but might have failed to bring to the attention of the readers in the submitted manuscript.
>
> $(i)$ We agree with the reviewer that our algorithm builds upon existing common privacy mechanisms, although several small contributions are required to make it as tight and as generally applicable as we want. For example, we do not require any known bounds on $w^\star$, $\Sigma$, and $\sigma^2$.  However, **we want to emphasize that most of our technical contribution is in the convergence analysis (that is orthogonal to privacy mechanisms and their guarantees).** To achieve the optimal error rate of $||\hat{w} -w^\star||_\Sigma = O(\alpha \sigma) $ with near optimal sample complexity of $n=\tilde{O}(d/\alpha^2 + \kappa^{1/ 2}d/(\varepsilon \alpha))$, we came up with a novel analysis technique in Appendix G steps 3 and 4.
> More precisely, naive linear regression analysis can only show a suboptimal error rate of $||\hat{w} -w^\star||_\Sigma = \tilde{O}(\kappa \alpha \sigma) $ with sample size $n=\tilde{O}(d/\alpha^2 + \kappa^{1/ 2}d/(\varepsilon \alpha))$. This follows from Eq (41):
>
> $$w_{t+1}-w^\star = (1-\eta\hat\Sigma) (w_t-w^\star)  + \eta u_t + \eta \phi_t \nu_t.$$
>
> From Eqs. (37), (38), (39), (40), and $\eta=1/\lambda_{\rm max}(\Sigma)$, it follows that
>
> $$ ||w_{t+1}-w^\star||_\Sigma \leq (1-\frac{1}{\kappa})||w_t-w^\star||_\Sigma + \alpha (\sigma+||w_t-w^\star||_\Sigma),$$
>
> where we omitted constants for simplicity, which after $T=\tilde O(\kappa)$ iterations achieves a **sub-optimal** error rate $||w_{T}-w^\star||_\Sigma = \tilde{O}(\kappa \alpha \sigma)$.
>
>
> One attempt to get around it is to take the Euclidean norm instead, which gives, after some calculations,
> $${\mathbb E}[ ||w_{t+1}-w^\star||^2]  \leq {\mathbb E}[ ||w_t-w^\star||^2] - \eta\Big( ||w_t-w^\star||_\Sigma^2 - \alpha^2\sigma^2 \Big) \;.$$
>
>
> This implies that ${\mathbb E}[||w_{t+1}-w^\star||^2]$ strictly decreases as long as $||w_t-w^\star||_\Sigma^2 > C \alpha^2\sigma^2$, which is the desired statistical error level we are targeting. With this analysis we can show that in  $T=\tilde O(\kappa)$ iterations, there exists at least one model $w_t$ that achieve ${\mathbb E}[||w_t-w^\star||_\Sigma^2] =\tilde O( \alpha^2\sigma^2)$ among all the intermediate models we have seen.
>
> However, the problem is that under differential privacy, there is no way we could select this good model $w_t$ among $T$ models that we have, as privacy preserving techniques for model selection are not accurate enough to achieve the desired level of accuracy. Hence, we came up with a novel analysis that does not suffer from such issues. Precisely, our analysis technique in Appendix G steps 3 and 4 opens up the iterative updates from beginning to end, and exploits the fact that $\lambda_{\rm max}((\eta\Sigma)^{1/ 2} (1-\eta \Sigma)^i(\eta \Sigma)^{1/ 2})$ is upper bounded by $1/(i+1)$. This gives the near optimal guarantee we provide in our paper. We believe this part of the analysis is novel and a significant technical contribution.
>
>
>
> $(ii)$ **Our label corruption for DP linear regression is an important problem at the intersection of robustness and privacy.** This is because this is the only setting that we know of where optimal sample complexity can be achieved with computationally efficient algorithms under sub-Gaussian data. For example, for a simpler problem of mean estimation of sub-Gaussian data with adversarial corruption and differential privacy, known best computationally efficient algorithms require strictly sub-optimal sample complexities of either $n=O(d^{3/ 2} / (\varepsilon \alpha))$ or $n=O(d/(\varepsilon \alpha^2))$ (citations [Liu et al. 2021] and [Hopkins et al. 2022] in our submitted version).  These have nontrivial gaps to the lower bound of $n=\Omega(d / (\varepsilon \alpha))$. So far, this can only be closed by computationally inefficient algorithms such as those in [Liu et al, 2022a] or [“Covariance-Aware Private Mean Estimation Without Private Covariance Estimation” by Gavin Brown, Marco Gaboardi, Adam Smith, Jonathan Ullman, and Lydia Zakynthinou.].  We have added these citations in the revision.
>
> A similar gap exists for DP linear regression also under sub-Gaussian data with adversarial corruption of covariates. In this paper, we have identified a non-trivial setting where near-optimality can be achieved with a computationally efficient algorithm, which is the **label-corruption setting** we consider in this paper. We believe providing a near optimal algorithm for such a case is an important contribution in the literature at the intersection of robustness and privacy.

---

> > ### Author Response · Authors · 2022-12-09
> > **Have we addressed your concerns?**
> >
> > We would like to thank the reviewer again for their constructive feedback. We tried our best to clarify the technical novelty of our analysis. The tight dependence on \kappa would not have been possible without the novel technical analysis. If there are any remaining concerns, we are happy to provide further information.

---

### Official Review · Reviewer_gS7K · 2022-10-23

**Confidence:** 5
**Correctness:** 3
**Technical Novelty And Significance:** 3
**Empirical Novelty And Significance:** 2
**Recommendation:** 6

**Clarity, Quality, Novelty And Reproducibility:**

I found the problem statement of the paper interesting and important. The authors nicely combine literatures from the DP linear regression and Robust linear regression to produce useful algorithms for the proposed setting. The authors have adequately cited related work as well as honestly commented on differences and similarities with existing literature. I think the paper is well-written. It would be good to bring the experiments in the main body. The submission looks technically sound and makes a solid theoretical contribution.  I didn't completely go through proofs but the intuition of breaking gradient update into contraction, noise from data and DP, bias introduced by clipping and corruption makes sense. I think the algorithms and the provided guarantees are important because the work can be applied in fields where adversary emerges and where there is a also a need of privacy and robustness like networks and finance.



**Strength And Weaknesses:**

Strength:
The paper is well-motivated since privacy and robustness become more and more important in ML model. This is the first efficient linear regression algorithm to provably guarantee both DP and robustness. The authors propose a private norm estimator and a robust private distance estimator by using private histogram mechanism for adaptive clipping and also give theoretical results for this algorithm. Based on this algorithm, they propose their DP-SGD method. The authors give both theoretical sample complexity and experimental results for the method. The proofs seem to be correct, and the experimental results also accord with their analysis.

Weaknesses:
1. The lower bound in Section 3.3 is not clear.
2.  The ‘n’ in equation (1) should be B_t.
3.  The step 1 of algorithm 1 should be three subsets.



**Summary Of The Paper:**

This paper studies the problem of linear regression under DP constraint and adversarial corruption. The authors propose a kind of variant of DP-SGD with a full-batch GD to improve sample complexity and adaptive clipping to guarantee robustness. Their theoretical results can improve the SOTA sample complexity without adversarial corruption. Under label corruption, they give the first provably linear regression algorithm to guarantee both DP and robustness. They also give experimental results on synthetic data to verify their theoretical results.



**Summary Of The Review:**

The lower bound is not clear.

---

> ### Author Response · Authors · 2022-11-12
> **Thank you for your constructive feedback**
>
> 1: **Our lower bound in Proposition 3.1 does not provide a lower bound on sample complexity (that follows from other lower bounds we cite) but provides a lower bound on the error rate one can achieve under adaptive adversarial corruption.** Our upper bound (Theorem 3) indicates that with sample size $n=\tilde O(d/\alpha^2+\kappa^{1/2}d\log(1/\delta)/(\varepsilon \alpha))$, our algorithm is able to achieve $\mathbb{E}[||w_T-w^\star||_{\Sigma}^2]=\tilde{O}(\sigma^2 \alpha^2)$ under $\alpha$ fraction of label corruption.
> Our lower bound in Proposition 3.1 indicates that even with infinite samples, no algorithm can achieve an error lower than $\Omega(\alpha^2\sigma^2)$.
>
> Precisely, $$\min_{\hat w, n}\max_{D_n \in D_{\sigma^2,\Sigma,\alpha}}\mathbb{E}[||\hat w-w^\star||_{\Sigma}^2]=\Omega (\sigma^2 \alpha^2)$$
>
> where the minimum is taken over all estimators $\hat w$ that take $n$ i.i.d samples with $\alpha$-corruption and we allow any number $n$ of samples including infinity, and the maximum is taken over all size $n$ datasets $D_n$ from sub-Gaussian distributions with covariate covariance $\Sigma$ and noise variance $\sigma^2$ and corrupted by an adaptive adversary who corrupts $\alpha$ fraction of the labels. Together with the upper bound, this lower bound shows that we achieve the best error rate one can hope for, while requiring minimal sample complexity; the second part of this claim follows from comparisons to other lower bounds in the literature, e.g.,  (Liu et al., 2022b, Corollary C.2) and  Bhatia et al. (2015); Dalalyan & Thompson (2019), as explained in Section 3.2.
> We have added a formal theorem statement for the lower bound in Section 3 of the revision.
>
>
> 2 and 3: These two are typos and we have fixed them and marked them as blue in the revision.
>
> Additionally, we would like to direct the reviewer’s attention to a detailed explanation of the novelty in the convergence proof in our response to reviewer p2zY.

---

> > ### Author Response · Authors · 2022-12-09
> > **Have we addressed your concerns?**
> >
> > We would like to thank the reviewer again for the insightful feedback. We have revised our lower bound to be more readable and clear. We believe the paper has improved significantly as a result. If there are any remaining concerns we are happy to address it further.

---

### Official Review · Reviewer_bBtg · 2022-10-24

**Confidence:** 3
**Correctness:** 3
**Technical Novelty And Significance:** 2
**Empirical Novelty And Significance:** 2
**Recommendation:** 5

**Clarity, Quality, Novelty And Reproducibility:**

The paper is clearly written, the quality of the empirical work seems fine but there is a lack of adequate theoretical developments. The algorithm may have some elements of novelty, and the results look reproducible.


**Strength And Weaknesses:**

Strengths:


Authors have a good grasp of some of the challenges of differentially private computations in the linear regression context. The paper is well-written. Section~4 on adaptive clipping is an interesting contribution, where I think the results can be improved, but the approach of using a a robust approach and trying to enforce differential privacy on it is interesting.


Weakness:

A few of the weaknesses are technical and can possibly be easily corrected. However, some of the weaknesses are probably not adjustable with minor changes.

1) The use of sub-Weibull distributions is more of a distraction. This is primarily because we would want the differentially private estimator to have consistency and other desirable statistical properties, for which sub-Gaussian conditions are often necessary. It is not clear what sub-Weibull adds to either the robustness or differential privacy components of this paper. Also, this assumption does not seem verifiable.

2) The requirement that $\bar{\alpha}$ is known in Assumption 2, and the complex bound for it seem unverifiable and not very practical.

3) For high-dimensional regression, it is not clear that the proposed scheme can achieve sparsistency.

4) Along the same lines, it is entirely possible that the norm or non-zero elements of a sparse $w^{*}$ may not have any relation with the leading eigen values/vectors of $\Sigma$. Hence it is not clear the bounds for the weighted loss function considered here relate to the sparsity of $w^{*}$ or the noise variance.

4) Also, it is not clear what additional complexity is arising from the corruption in the labels arising from an adversarial framework. This would only make sense if an oracle knew the original observations (indexed by $S_{r}$), and the proposed algorithm achieved some kind of a risk bound with respect to this oracle. Since the corruption can be arbitrary, it seems a standard robust statistics approach would have been sufficient here.



**Summary Of The Paper:**

Authors study linear regression from a differential privacy perspective. They propose a variant of the differentially private stochastic gradient descent (DP-SGD) algorithm with two innovations: a full-batch gradient descent to improve sample complexity and an adaptive clipping to guarantee robustness. Some theoretical results and empirical evidence are presented.


**Summary Of The Review:**

This is an interesting paper overall, but when we dig a little deeper there are several unanswered questions.

---

> ### Author Response · Authors · 2022-11-12
> **Thank you for your insightful feedback**
>
> We address each comment as follows.
>
> 1. We agree that the subWeibull assumption is not verifiable from given samples. **We agree with the reviewer that the main contribution of our work can be presented under the simpler sub-Gaussian assumption.** We chose to present the more detailed dependence on the tail of subWeibull distributions to $(i)$ be consistent with the most relevant prior work of (Varshney et al 2022) and $(ii)$ show the dependence in the tail parameter $a$. Another possible way of presenting our work is to present everything for Sub-Gaussian first, mainly to simplify the assumption and notations, and then have one section which generalizes the results to Sub-Weibull. For example, the informal versions of our theorems in Section 1 can be presented under sub-Gaussian assumptions. We want to emphasize that further simplifying the assumption to be **strictly Gaussian** would lose important aspects of the problem (as opposed to many differential private estimation literature, e.g., (Karwa and Vadhan,2017) and (Ashtiani, Liaw, 2021), which assumes Gaussianity without any loss). This is explained at the beginning of Section 3.1. Concretely, under Gaussian assumptions, the squared norm of the covariate is lower bounded by, say, ${\rm Tr}(\Sigma)/2$ such that we do not need to separately clip the residual and the covariate. We can use a single clipping by the norm of the gradient, which is commonly used. Only under sub-Gaussian, or more general sub-Weibull, distributions, the (adaptive and label corrupting) adversary has more power and correspondingly we require two separate clipping algorithms.
>
>
> 2. **Knowing an upper bound on the fraction of corruption (such as our $\bar{\alpha}$) has been a standard assumption made in many recent papers on robust estimation such as Bhatia et al. 2015 and Diakonikolas et al. 2017.** Very recently there is some progress in reducing this requirement, also in a meta-algorithm way:  “Robust estimation algorithms don’t need to know the corruption level.” by Jain, Orlitsky, Ravindrakumar. The idea of their meta-algorithm is to start from running an algorithm with $\bar{\alpha}=1$ and decreasing the value of $\bar{\alpha}$ by a factor of 2 at every iteration. This will only incur an additional $\log(1/\alpha)$ multiplicative factor in the sample complexity of our result due to private composition, but removing the need of knowing $\bar{\alpha}$. This is an exciting research direction but outside the scope of this paper.
>
> 3. **We did not intend to solve sparse linear regression in this paper, and hence our algorithm is not tailored for sparsity.** Therefore, we agree with the reviewer that our algorithm does not achieve sparsistency as is. We want to first note that even for non-DP linear regression, algorithms that are not explicitly enforcing sparse solutions will not guarantee sparsistency, as far as we know.
> If $d$ is fixed and $n$ increases, our algorithm achieves zero error eventually and hence recovers the support. However, in the high-dimensional setting where $d$ also increases with $n$, special algorithms are required. There have been a lot of works making progress in this sparse linear regression setting, for example, Cai et al. (2019), Kifer et al. (2012), and recent papers “Differentially Private Iterative Gradient Hard Thresholding for Sparse Learning” by Wang and Gu (2019), and “High Dimensional Differentially Private Stochastic Optimization with Heavy-tailed Data” by Hu,  Ni, Xiao, and Wang (2021). In the sparse setting, we could potentially apply a hard thresholding step after each iteration similarly as in Cai et al (2019), Gu et al  (2019) and Wang et al (2021), and get sparsistency in the final guarantee. We think this is beyond the scope of this paper and may be an interesting avenue for future work.  We have added all these citations in the revision.
>
> Continued in the next response

---

> > ### Author Response · Authors · 2022-11-12
> > **Continued**
> >
> > 4. **$\Sigma$-norm is a standard measure of error for linear regression (including sparse linear regression) for the following reasons.** First of all, this corresponds to the prediction error of the problem on new incoming samples from the same distribution for the natural squared loss. Let $(x,y)$ be the new sample from the same distribution. The prediction error is defined as $\mathbb{E}_{(x,y)}[(\hat w^T x - y)^2]=\mathbb{E}[( (\hat{w}-w^\star)^T x - z)^2]=||\hat w - w^\star||^2_\Sigma+ \sigma^2$, where $y=x^Tw^\star+z$. Hence, $\Sigma$-norm is a natural choice for the error metric.
> > Secondly, this adjusts the scale of error in each direction according to the signal-to-noise ratio in that direction. Consider a linear regression problem in $(x,y)$ from $y=<x,w^*>+z$ with covariance $\Sigma$ of $x$. The right way to measure the error is to consider the error when the covariance is identity. Hence, define $\tilde{x} = \Sigma^{-1/ 2} x$ such that covariance of $\tilde x$ is identity. Then we can imagine solving the same problem, but given $(\tilde{x},y)$ from $y=<\tilde x,\tilde w^\star>+z$ where $\tilde w^\star = \Sigma^{1/ 2} w^\star $. This is the same problem, just that we observe whitened covariates with identity covariance. In this case, there is no ambiguity on what the right measure is for the error: $||\hat{\hat w} - \tilde w^\star||$. Translating this right measure of error to the original problem, we get that $|| \Sigma^{1/ 2} \hat w - \Sigma^{1/ 2} w^\star|| = ||\hat w - w^\star ||_\Sigma$ is the right measure, where $\hat{\hat w}=\Sigma^{1/2}\hat{w}$.
> > The following papers use $\Sigma$-norm to measure the error achieved by sparse linear regression algorithms: Suggala et al (2019), Kifer et al (2012), Cai et al (2019), and Wang et al (2021).
> > If there are remaining concerns with the use of $\Sigma$-norm, we are happy to address them further in subsequent discussions.
> >
> > 5. **In label-corruption settings, there is a significant difference between adaptive adversary and oblivious adversary.** In our paper, we consider the **adaptive label corruption** model where the adversary is able to inspect the data and then corrupt $\alpha$-fraction of the labels. In this case, we are able to achieve the information-theoretic lower bound of the error $\mathbb{E}[||w^*-\hat{w}||_\Sigma^2]=\Omega(\sigma^2\alpha^2)$ (See Section 3.3 for a lower bound) with optimal sample complexity of $n=\tilde{O}(d/\alpha^2 + d/(\alpha \varepsilon))$.\
> > There is also a weaker notion of the adversarial model, i.e. **oblivious label corruption** model, where the adversary corrupts labels in complete ignorance of the uncorrupted data. In the oblivious model, efficient robust algorithms achieve zero error as sample size increases to infinity for all $\alpha<1/2$ (Bhatia et al., 2017; Suggala et al., 2019). Such a fundamental difference between adaptive corruption and oblivious corruption does not exist when both the covariate and the label are corrupted. This gap only exists for the label corruption setting we study. This is the additional complexity in the problem that arises from adversarial corruption we assume. Namely, achieving the fundamental limit of the error $\mathbb{E}[||w^*-\hat{w}||_\Sigma^2]=\tilde O(\sigma^2\alpha^2)$ with optimal sample complexity. This is not an easy task and we would like to direct the reviewer’s attention to a detailed explanation of the novelty in the convergence proof in our response to reviewer p2zY.
> > In the oblivious label corruption setting, it might be possible to achieve a risk bound with respect to the oracle who knows the original data as the reviewer pointed out. This has not been studied under DP, as far as we know, and we believe that is outside the scope of this paper. \
> > Applying standard robust statistics algorithms directly to a Robust and DP problem inevitably increases the sample complexity or computational complexity. If we give up **computational efficiency**, then there are a few standard robust statistics approaches for linear regression that could be potentially applied to our problem. Regression depth based approaches like Liu et al., 2022a achieves optimal error with optimal sample complexity but requires exponential time complexities (analysis of regression depth in non-DP setting is presented in Chao Gao., 2017). Otherwise, efficient approaches suffer from strictly sub-optimal sample complexity under even simpler problems of mean estimation, e.g., Liu et al. 2021. Even for mean estimation, whether we can achieve optimal error rate using $O(d)$ samples in polynomial time is still an open problem. Our results demonstrate that when only labels are corrupted, we can achieve near optimality for linear regression. We would like to direct reviewer’s attention to our response to reviewer p2zY for a more detailed explanation of why private linear regression under label corruption is special.

---

> > > ### Author Response · Authors · 2022-12-09
> > > **Have we addressed your concerns?**
> > >
> > > We would like to thank the reviewer again for their constructive feedback. We have tried our best to address the comments, and we believe we have clarified some important points raised by the reviewer. We are happy to provide further information if there are any remaining concerns.

---

### Official Review · Reviewer_gANM · 2022-10-28

**Confidence:** 3
**Correctness:** 2
**Technical Novelty And Significance:** 2
**Empirical Novelty And Significance:** 3
**Recommendation:** 5

**Clarity, Quality, Novelty And Reproducibility:**

The presentation is OK, and the proposed method looks novel, but it needs further clarification on the improvements.

**Strength And Weaknesses:**

The strength of the paper:
1. The proposed method is able to handle corrupted labels.
2. Experiments validate the effectiveness of the proposed method.

The weaknesses of the paper:
1. In the noncorrupted setting, it is unclear to me why it can improve the sample complexity requirement compared with Varshney et al. 2022. For the minibatch method proposed by Varshney et al. 2022, it seems that the minibatch size is chosen in the order of the full batch size. Thus it is unclear to me why using the full gradient as suggested in this paper can improve the sample complexity? Whether the improvement comes from Algorithm 2? These need to be further clarified.
2. The requirements on $S_1$, $S_2$ and $S_3$ are not clear.
3. Why the requirement of $n$ in equation (5) is independent of $\kappa$? According to the last inequality on Page 16, it seems that the $\bar \alpha$ depends on $\kappa$, and thus $n$ should also depend on $\kappa$.
4. Although the proposed method is able to handle label corruption, the corruption level is so small that it should be less than the desired accuracy level. Can you relax this requirement?

**Summary Of The Paper:**

This paper studies the problem of differentially private linear regression with potentially corrupted labels. To handle the corrupted labels, the authors proposed a new private adaptive clipping technique. The proposed method can deal with both privacy concerns and corrupted labels. The experiment results validate the effectiveness of the proposed method in the corrupted setting. Furthermore, the authors claim that the proposed method can also improve the sample complexity requirements to achieve a certain level of accuracy in the noncorrupted case compared with existing methods.

**Summary Of The Review:**

The author proposed an efficient, robust, and differentially private algorithm for solving the linear regression problem. The proposed method seems to be promising. However, it suffers from the low corruption level and the improvements over the previous method in the noncorrupted setting are unclear.

---

> ### Author Response · Authors · 2022-11-12
> **Thank you for the constructive feedback**
>
> 1. **(Varshney et al 2022) uses a batch size of $O(n/\kappa)$, which results in a sensitivity that is larger by a factor of $\kappa$ compared to our full-batch algorithm. They consequently pay in the larger noise added for privacy.** More precisely, (Varshney et al 2022) use a streaming algorithm with $O(\kappa \log n)$ iterations, which results in a per-iteration batch size of $O(n/(\kappa\log n))$. The resulting sensitivity of the average of the gradients is $\tilde O(\kappa \cdot \text{clipping threshold} /n )$. For our approach, since we use a full batch with batch size $n$, the sensitivity is $O( \text{clipping threshold} /n )$. So, we add a smaller noise for privacy compared to Varshney et al 2022, and consequently obtain better dependence of $\kappa$ in sample complexity. To summarize, our improvements in the sample complexity in its dependence in $\kappa$ mainly come from 1) using the full batch gradient descent, and 2) the novel convergence analysis. We would like to direct the reviewer’s attention to a detailed explanation of the novelty in the convergence proof in our response to reviewer p2zY.  \
> **Regarding Algorithm 2: While Algorithm 2 doesn’t directly improve the sample complexity, we make a couple of important contributions in the algorithm.** It is similar to DP-STAT (Algorithm 3 in Varshney et al 2022) for estimating $||w-w^{\star}||_\Sigma+\sigma $, but with three important improvements: 1) DP-STAT requires the knowledge of the domain size, which in turn requires the knowledge of $||w^\star||_\Sigma+\sigma$. Whereas, Algorithm 2 in our paper doesn’t require the knowledge of domain size. 2) Our utility guarantee improves the $K$ and $\log^{2a}(n)$ dependence. 3) Our estimator is robust against label corruption. Note that using DP-STAT with our algorithm would not change the sample complexity and its dependence on $\kappa$, but we would need to know $||w^\star||_\Sigma+\sigma$ and also need to give up robustness against label corruption. We have clarified these differences in Remark 4.1 of the revision.
>
> 2. **For notational convenience we let $|S_1|=|S_2|=|S_3|$ and merge all sufficient conditions on $S_1$, $S_2$, and $S_3$ together.** We assume that the dataset of size $3n$ is partitioned into three equal-sized sets $S_1,S_2,S_3$, each of size $n$. This choice makes the notations much simpler for presentation, while not changing the final guarantee as we do not keep track of the constant factors in sample complexity. It is true that the sufficient conditions on the sizes of $S_1, S_2$, and $S_3$ are different, and one could potentially gain in the constant factor in sample complexity from better partitioning of the dataset. In particular, it is sufficient to have $|S_1|=\tilde O(\log^{2a}(1/\delta)/\varepsilon)$, $|S_2|=\tilde O(d/(\bar\alpha \varepsilon))$ and $|S_3|=\tilde O(d/\alpha^2+\kappa^{1/2}d\log(1/\delta)/(\varepsilon \alpha))$, with large enough constants.
>
>
> 3. **$\bar{\alpha}$ in Theorem 4 doesn’t depend on $\kappa$.** We require that $\bar{\alpha}$ satisfy $37C_2K^2\bar{\alpha}\log^{2a}(1/(6\bar{\alpha}))\leq 1/4$, where $C_2$ is a universal constant, $K$ and $a$ are the distribution parameters for sub-Weibull distributions. We have written these conditions explicitly within Theorem 4.
> **Why $n$ is independent of $\kappa$ : Note that in Theorem 4, we only need to estimate distance up to a  constant multiplicative error, as opposed to an error that depends on our final end-to-end desired level $\alpha$. Consequently, we require smaller sample complexity (that doesn’t depend on $\kappa$) than other parts of our approach.**
> The sample complexity only depends on $K$ and $a$ and not on $\kappa$. For example, for sub-Gaussian distributions, $\bar{\alpha}$ is a constant in $(0,1)$, which does not depend on $\kappa$. This is because the utility guarantees we need in this private distance estimator is a constant multiplicative error to the true distance $||w^\star-w||_\Sigma+\sigma$, independent of our final desired accuracy. Hence, the guarantee of the private distance estimator in Theorem 4 Equation (5) is different from other guarantees, in the sense that $n$ depends on $\bar{\alpha}$ instead of the desired accuracy $\alpha$.
> In the last equation on Page 16, we need to show the corruption level of each partition is less than $2\bar{\alpha}$, which would require $n\gtrsim k/\bar{\alpha}$. Here this $k$ is the number of partitions (defined in Algorithm 2), which is independent of $\kappa$.
>
> Continued in the next response.

---

> > ### Author Response · Authors · 2022-11-12
> > **Continued**
> >
> > 4. **The requirement that the final error should be larger than the corruption level is fundamental and cannot be relaxed under the strong adaptive label adversary we assume in our paper.** In the **oblivious** label corruption model (without privacy), where the adversary corrupts $\alpha$-fraction of the labels in complete ignorance of the data, even if the corruption level is a large constant, it is possible to achieve a small error (Bhatia et al., 2017; Suggala et al., 2019). However, for the **adaptive** label corruption model that we assume in this paper, there is a fundamental lower bound showing that one cannot achieve an error smaller than $\Omega(\alpha)$, when $\alpha$-fraction of labels are corrupted adaptively. One example of such an information-theoretic lower bound is our Proposition 3.1 (a similar lower bound can be found in (Bakshi&Prasad, 2021). From these results in robust statistics, we know that the corruption level can be only smaller than the desired error level (actually, the corruption level is given and the error level must be larger than the corruption level). The main contribution of our paper is that we can match this fundamental limit with near-optimal sample complexity while guaranteeing differential privacy simultaneously. Hence, the fact that we cannot have more corruption than our target error is fundamental and cannot be improved upon (no matter what algorithm you use).

---

### Decision · Program_Chairs · 2023-01-20

**Decision:**

Reject

**Justification For Why Not Higher Score:**

The paper has a few technical issues and does not merit acceptance

**Justification For Why Not Lower Score:**

N/A

**Metareview: Summary, Strengths And Weaknesses:**

In this work, the authors study the problem of linear regression under $(\varepsilon,\delta)$-differential privacy when a fraction of response variables are adversarially corrupted. They propose a variant of the differentially private stochastic gradient descent (DP-SGD) algorithm that uses a full-batch gradient descent along with an adaptive clipping to guarantee robustness. When there is no adversarial corruption, the algorithm achieves near-optimal sample complexity and improves upon the existing state-of-the-art approach. The authors claim that under label corruption, it is the first efficient linear regression algorithm to provably guarantee both $(\epsilon,\delta)$-DP and robustness.

The reviewers liked the fact that the proposed method is able to handle corrupted labels and thought the experiments somewhat validated the effectiveness of the method. The reviewers raised a variety of issues including. (1) lack of clarity that sample complexity can be improved compared to Varshney et al in the non-corrupted setting, (2) equation (5) being a restrictive requirement (3) corruption level too restrictive (3) impracticality of Assumption 2. The authors provided a thorough response. This response however does not address all of the issues raised. My own reading is that this paper studies an interesting problem but in its current form is not ready for prime time and therefore I can not recommend acceptance.